# Charge-separation driven mechanism via acylium ion intermediate migration during catalytic carbonylation in mordenite zeolite

Wei Chen [1], Karolina A. Tarach [2], Xianfeng Yi [1], Zhiqiang Liu [1], Xiaomin Tang[1], Kinga Góra-Marek [2] ✉ & Anmin Zheng [1,3] ✉

By employing ab initio molecular dynamic simulations, solid-state NMR spectroscopy, and two-dimensional correlation analysis of rapid scan Fourier transform infrared spectroscopy data, a new pathway is proposed for the formation of methyl acetate (MA) via the acylium ion (i.e., $CH_3 - C \equiv O^+$) in 12-membered ring (MR) channel of mordenite by an integrated reaction/diffusion kinetics model, and this route is kinetically and thermodynamically more favorable than the traditional viewpoint in 8MR channel. From perspective of the complete catalytic cycle, the separation of these two reaction zones, i.e., the C-C bond coupling in 8MR channel and MA formation in 12MR channel, effectively avoids aggregation of highly active acetyl species or ketene, thereby reducing undesired carbon deposit production. The synergistic effect of different channels appears to account for the high carbonylation activity in mordenite that has thus far not been fully explained, and this paradigm may rationalize the observed catalytic activity of other reactions.

The unique pore architectures and active acidic sites of zeolites, coupled with the spatial confinement afforded by the framework structure, enables zeolites to function as unique shape-selective catalysts for the direct formation of specific chemical products[1]. Carbonylation in zeolites provides a convenient route and platform in both zeolite catalysts and multi-functional catalysts to functionalize C1 intermediates via formation of carbon-carbon bonds at relatively low temperatures to realize syngas conversion[2], $CO_2$ reduction[3], and methanol to olefins[4]. As a typical zeolite catalyzed reaction mediated by shape-selection constraints, the carbonylation of methanol (MeOH) or dimethyl ether (DME) with carbon monoxide (CO) is a cost-effective process for the production of methyl acetate (MA) or acetic acid (AcOH)[5–7]. Mordenite containing an 8-membered ring (8MR) channels has found great success for the carbonylation reaction and exhibits high activity[8–10]. The Brønsted acid sites (BAS) in the 8MR channels and the appropriate pore shape and size guaranteeing the unique chemical/physical interactions between the pore walls and the reaction intermediates have been found to be the decisive factors responsible for the high carbonylation efficiency. The generally accepted rate-determining step of carbonylation, that is, formation of acetyl intermediates via C–C bond coupling between surface methyl species (SMS) and CO, has been broadly theoretically investigated[11–13], and the fact that C–C bond coupling in the 8MR channel is kinetically more favorable than that in the 12-membered ring (12MR) channel, with a free energy barrier difference of ca. 25 kJ/mol. Moreover, the presence of BAS in 12MR channel of mordenite (MOR-12MR) is thought to favor C-C bond coupling leading to undesired coke formation and eventually catalyst deactivation. Hence, attempts were made at deactivation of these BAS, most notably by (i) pyridine pre-adsorption[14–16], (ii) template-induced and post-treatment, forming 12MR channels with smaller Al distributions[17–19], (iii) so-called bottom-up route to synthesize nano-sized mordenite as well as a top-down approach to engineer hierarchical structures of mordenite by such methods as acid, base, and steam treatments[20,21]. In addition, many other factors of zeolite can

[1]State Key Laboratory of Magnetic Resonance and Atomic and Molecular Physics, National Center for Magnetic Resonance in Wuhan, Innovation Academy for Precision Measurement Science and Technology, Chinese Academy of Sciences, Wuhan 430071, P. R. China. [2]Faculty of Chemistry, Jagiellonian University in Krakow, Gronostajowa 2, 30-387 Krakow, Poland. [3]University of Chinese Academy of Sciences, Beijing 100049, P. R. China.
✉ e-mail: kinga.gora-marek@uj.edu.pl; zhenganm@wipm.ac.cn

also greatly influence the catalytic performance of carbonylation, like membrane size and thickness, acidic site density, surface barriers, etc. In fact, both of these experimental approaches aim at minimizing C−C bond coupling in the 12MR channel in favor of carbonylation and MA formation in the 8MR channel by shortening the diffusion pathway and alleviating the diffusion resistance in the 12MR channel. However, these studies underscore the importance of the 12MR channel as a transport channel for reagents and products, suggesting that its role in zeolite mediated carbonylation has been overlooked in favor of the prominent role played by 8MR channel[22].

More importantly, MA formation between surface acetyl and MeOH/DME as the following step of C−C bond coupling can be hindered by the size mismatch of the appropriate reactants and the narrow pores in the 8MR channel and the side pocket in mordenite (Fig. 1a). The complete catalytic cycle of DME carbonylation in the 8MR channel of mordenite is shown in Fig. 1b. Although MA formation via surface acetyl and MeOH in mordenite is feasible (Supplementary Fig. 1), the large size of the DME substrate requires an immense free energy barrier of 249.6 kJ/mol to reach the appropriate transition state (TS3, Fig. 1c) that overcomes the strong steric hindrance resulting from

entrance of DME into the narrow 8MR channel and side pocket (Supplementary Fig. 2). Detailedly, the energy decomposition analysis to host-guest interaction between reacting species in the MOR-8MR and MOR-12MR for TS3 further confirms the much higher steric strain present in the narrower 8MR channels as revealed by Pauli replusion interaction ($\Delta E_{Pauli}$) (Fig. 1d), as well as the reduced density gradient surface in Fig. 1e, f. Notably, although the free energy barrier in side pocket is greatly decreased relative to that in 8MR channel, this free energy barrier (164.1 kJ/mol) of MA formation in side pocket is greatly larger than the free energy barrier (87.0 kJ/mol) of C−C bond coupling in 8MR channel (Fig. 1c). Herein, the free energy barrier of MA formation in side pocket is contributed by the unfavorable adsorption free energy (64.3 kJ/mol) and intrinsic activation energy (99.8 kJ/mol), which indicated that the side pocket with the larger space than 8MR channel is still not enough to allow the MA formation (Supplementary Fig. 2). MA formation in MOR-8MR via reaction of DME and surface acetyl should be strongly inhibited in comparison to reaction in MOR-12MR, as evidenced by the difference on maximum free energy span ($\delta E$) (70.2 kJ/mol; Fig. 1c). However, the C−C bond coupling in MOR-12MR is theoretically infavorable[11–13], and catalytic experiments by

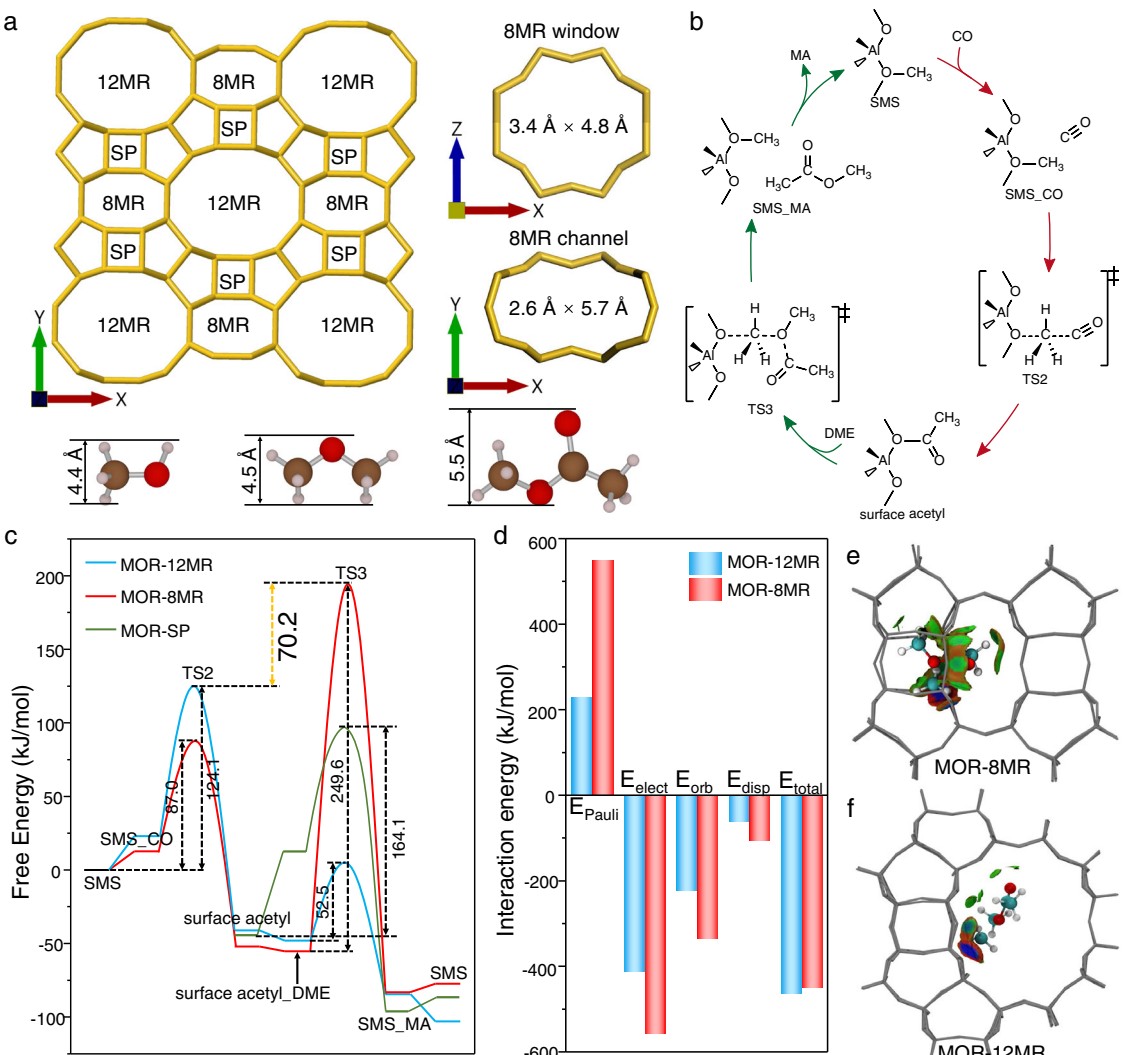

**Fig. 1 | Mechanism of DME carbonylation in mordenite. a** Pore and channel sizes of mordenite, kinetic sizes of methanol, dimethyl ether, and methyl acetate. The 8MR window refers to the opening between the 12MR channel and the 8MR side pocket (SP) of mordenite. **b** Catalytic cycle of DME carbonylation, the red arrows indicate the rate-determining step, and the green arrows indicate the fast step. **c** Free energy profile over MOR-8MR, MOR-12MR, and MOR-SP at 473 K obtained using the PBE-D3/dgdzvp method. **d** Energy decomposition analysis between organic species and zeolite framework at TS3 in MOR-8MR and MOR-12MR. Host-guest interactions of TS3 in the zeolite framework visualized using the reduced density gradient: **e** MOR-8MR, including both 8MR and SP; and **f** MOR-12MR. Red: repulsion, green: attraction.

Rasmussen et al.[13] appears to run counter to these purely theoretical considerations and therefore other factors must also be considered.

To resolve this contradiction, an alternative explanation is emerging that focuses on alternative pathways for MA formation that avoid the strong steric demands imposed on the reactants by the zeolite channel sizes. Consequently, ketene or the acylium ion (two derivatives of surface acetyl) are two plausible intermediates involved in MA formation due to their higher mobilities than surface acetyl[23]. Indeed, while Plessow et al.[4,24] confirmed the high activity of ketene and its analogues in transformations of methanol to olefins, but a large amount of ketene formation resulted in deactivation of the catalyst through the formation of coke precursors[25,26]. Recently, the acylium ion ($CH_3-C\equiv O^+$) formed via C−C bond coupling between SMS and CO in MOR-8MR was proven to be an alternative intermediate confined in the side pocket of mordenite[27], which should theoretically allow for highly efficient MA formation from MeOH and DME while avoiding the C-O bond rupture associated with the key intermediate of the catalytic cycle involving the traditional surface acetyl moiety. Moreover, previous studies that have confirmed the high mobility of cations with the high activity in zeolite structures, including the hydronium ion[28,29] and $Cu(NH_3)_2^+$ species[30,31], provide an interesting clue to rationalize the importance of the acylium ion as a key intermediate in DME carbonylation. In this context, it is highly probable that an acylium ion confined in the side pocket of mordenite migrates into 12MR channels to directly react directly with MeOH or DME.

To confirm the validity of these hypotheses on the mechanism of MA formation in mordenite, ab initio molecular dynamic (AIMD) simulations were carried out to explore the complete free energy profile of MA formation using surface acetyl and acylium ion as intermediates, while tantamount considering relocation of reactants and products between the 8MR and 12MR channels in mordenite from the viewpoint of integrated reaction/diffusion kinetics model, commonly disregarded in the literature reports. The dynamic evolutions and the temporal and spatial correlations of the diagnostic bands in the Fourier transform infrared (FTIR) spectra collected during the carbonylation reaction provided evidence for the formulated predictions.

## Results

### Methyl acetate formation via surface acetyl

To understand the commonly accepted pathway of MA formation via surface acetyl and DME/MeOH, the reaction pathway in the large MOR-12MR was inferred from the metadynamics (MTD)-AIMD simulations. MeOH is notably the more reactive species due to the lower free energy barriers (ΔF) than DME. In the presence of MeOH, interaction with surface acetyl forms MA via C-O bond rupture of surface acetyl with concomitant formation of a new C-O bond between the carbonyl group of the newly decoupled acetyl moiety and MeOH oxygen (Fig. 2a and Supplementary Movie 1). In contrast, the positively charged MAMe$^+$ (III) is a local minimum on the free energy profile because the demethylation of MAMe$^+$ (III) to MA (IV) is the energy-intensive step in MA formation between DME and surface acetyl (Fig. 2b and Supplementary Movie 2). However, if the adsorption free energies of DME and MeOH are taken into consideration (Supplementary Table 1), the maximum free energy span (δE) of MA formation is considerably increased, DME: 165.5 kJ/mol, MeOH: 138.7 kJ/mol. Based on these results, the rate determining step of the complete DME carbonylation process would inconsequently change from C−C bond coupling to MA formation due to the lower δE of the C−C bond coupling step (MOR-12MR: 155.7 kJ/mol)[27]. Therefore, reaction with DME to form MA in zeolites with large cavities is still an energy intensive process.

Although MeOH is predicted to be the ideal reactant to form MA, industrial processes involving zeolite catalytic MeOH carbonylation to MA are hampered by low catalytic activity and lifespan. The poor catalytic performance with MeOH observed in practice is due to the

formation of equimolar amounts of MA and water (Eq. (1)), while the formed water molecules are not released easily from the zeolite structure and hence hinder zeolite activity by formation of strong interactions with the BAS. Thus, MeOH dehydration ultimately inhibits the formation of SMS[32,33].

$$2CH_3OH + CO \rightarrow CH_3COOCH_3 + H_2O. \tag{1}$$

Based on the high ΔF of MA formation in MOR-12MR, the low activities and rapid deactivations of DME carbonylation in MOR-12MR should be attributed to the fact that the formed surface acetyl cannot rapidly transform to MA even in the presence of large number of DME molecules unless MeOH approaches it. Still, the accumulated surface acetyl can easily deprotonate to ketene and further dimerize to diketene as the precursor of carbon deposition. Therefore, there is a significant enhancement in the activity and life-span of mordenite catalysts when attempts are made at eliminating the BAS in 12MR channels[14–19].

Based on the above-described mechanism of MA formation in MOR-12MR, the mechanistic differences of MA formation via surface acetyl in MOR-8MR are worth investigating. First, DME was excluded from the reactant to form MA with surface acetyl due to the ultra-high barrier in Fig. 1c, and only MeOH as the sole reactant to react with surface acetyl in MOR-8MR to obtain the free energy profile at reaction temperature (473 K) by MTD-AIMD simulations (Fig. 2c and Supplementary Movie 3). In contrast to the obtained barrierless process by the static density functional theory (DFT) calculations in Fig. 1c, the free energy barrier in AIMD simulations predicting MA formation via surface acetyl and MeOH is much smaller at 62.0 kJ/mol since full consideration is given to the flexibility of the framework and the reactants at the reaction temperature[34–36]. Different from the concerted mechanism in MOR-12MR, MA formation in MOR-8MR is a stepwise one as displayed in Fig. 2d, e. The first step consists in the C−O bond rupture of surface acetyl to rapidly form an acylium ion, followed by C−O bond coupling by reaction of acylium ion and MeOH, with concomitant deprotonation of MAH$^+$.

Besides the reaction process, the diffusion processes of both reactant and product in narrow MOR-8MR were also considered as the important part of the complete reaction process by the umbrella sampling (US)-AIMD simulations (Supplementary Figs. 4–11). The diffusion of MeOH from the 12MR channel to the 8MR channel in mordenite need overcome a small diffusing free energy barrier (ΔF$_d$ = 29.6 kJ/mol) in conjunction with a higher free energy of 10.0 kJ/mol (Supplementary Fig. 8). However, the transport of MA formed in the 8MR channel into the 12MR channel requires a high ΔF$_d$ of 105.4 kJ/mol and an endothermic free energy of 51.1 kJ/mol to be overcome (Supplementary Fig. 10). This high ΔF$_d$ is related to the strong adsorption energy of MOR-8MR to MA including the hydrogen bonding interaction between BAS and MA and the van der Waals interactions between the MOR-8MR framework and MA (Supplementary Fig. 12). Considering MeOH adsorption in the MOR-12MR (Supplementary Table 1) as one part of the complete catalytic process that converts surface acetyl + MeOH to MA + BAS in mordenite, the maximum free energy span (δE) on the free energy profile reached 142.3 kJ/mol (Fig. 3). Considering such a large δE even larger than C−C bond coupling in MOR-8MR (128.6 kJ/mol), MA formation via reaction of surface acetyl and MeOH in MOR-8MR is an energetically unfavorable route from the perspective of integrated reaction/diffusion kinetics model.

### MA formation via migrated acylium ion

One derivate of surface acetyl, acylium ion in MOR-8MR was confirmed in our previous studies[27]. However, another derivative, ketene, is more rapidly protonated to form acylium ion in MOR-8MR as a kinetically and thermodynamically more favorable process (Supplementary

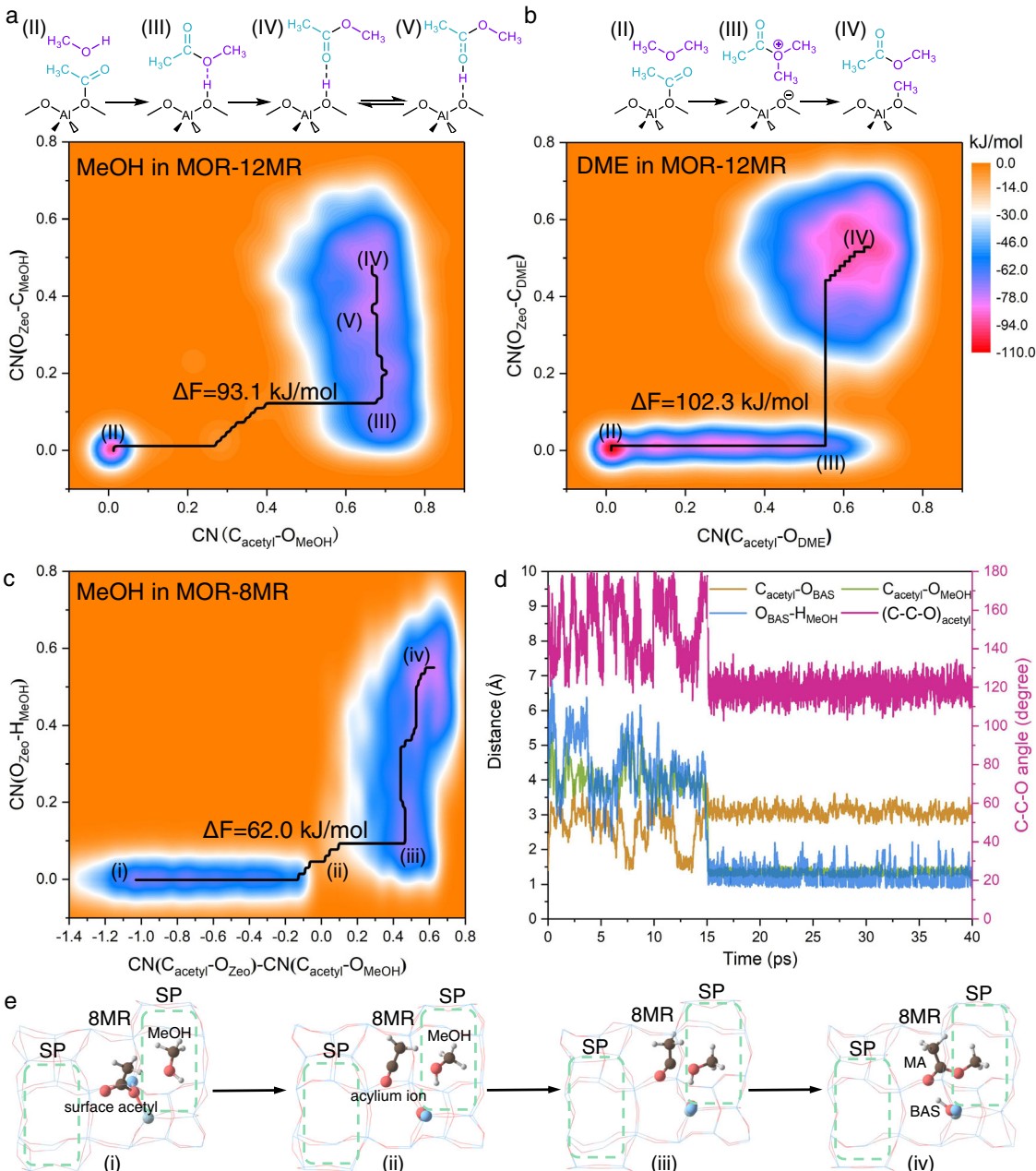

**Fig. 2 | Free energy profiles (473 K) and reaction routes of MA formation.**
**a** MeOH + surface acetyl and **b** DME + surface acetyl in MOR-12MR,
**c** MeOH + surface acetyl in MOR-8MR. **d** Dynamic evolution of key bond distance
and angle and **e** reaction process of **c**. SP: side pocket. The structures of (i)–(iv) in
**e** are corresponding to the local minimum of (i)–(iv) in **c**. The value of ΔF in **a**–**c** is
free energy barrier in the minimum free energy pathway. CN($C_{acetyl}$-$O_{MeOH}$):

coordination number between acetyl carbon and oxygen of MeOH, CN($C_{acetyl}$-
$O_{DME}$): coordination number between acetyl carbon and oxygen of DME, CN($O_{zeo}$-
$H_{MeOH}$): coordination number between oxygen of AlO$_4$ in zeolite and hydroxyl
hydrogen of MeOH. CN($O_{zeo}$-$H_{DME}$): coordination number between oxygen of AlO$_4$
in zeolite and carbon of DME. More motivation details of CN setting can be found in
Supplementary Methods and Supplementary Figs. 2, 3.

Fig. 13). Therefore, acylium ion was the sole alternative intermediate in
MOR-8MR, but MA formation would require the migration of the
acylium ion from the side pocket to the 12MR channel because steric
constraints in the narrow 8MR channel and side pocket prevent
the direct formation of MA via reaction of the acylium ion and
DME/MeOH.

To verify this hypothesis, we employed the US-AIMD simulations
to monitor the motion of the acylium ion in the presence and absence
of DME/MeOH in the 12MR channel (Fig. 4, Supplementary Fig. 14, and
Supplementary Movie 4). Notably, the positive/negative charge
separation between the acylium ion and Si-O⁻-Al leads to ΔF$_d$ of 57.9 kJ/
mol for the migration from the 8MR to the 12MR channels, with a free

energy compensation of 30.8 kJ/mol. Accordingly, this migration is
unspontaneous and thermodynamically unfavorable (Fig. 4a). How-
ever, under the long-distance attraction (*ca.* 5 Å) by DME or MeOH in
the 12MR channel of MOR, the mobility turned out to be kinetically and
thermodynamically conducive, and the respective ΔF$_d$ decreased to
47.9 kJ/mol (DME in Fig. 4b) and 38.8 kJ/mol (MeOH in Fig. 4c). Nota-
bly, the confinement effect of the side pocket plays a paramount role in
stabilizing acylium ion in relatively low energy state, and further
facilitating the mobility of acylium ion, as demonstrated by the sig-
nificantly higher energy required for transport of the acylium ion from
8MR to 12MR observed in MOR-8MR cluster models of decreasing size
(Supplementary Fig. 15). Besides, the presence of additional BAS in the

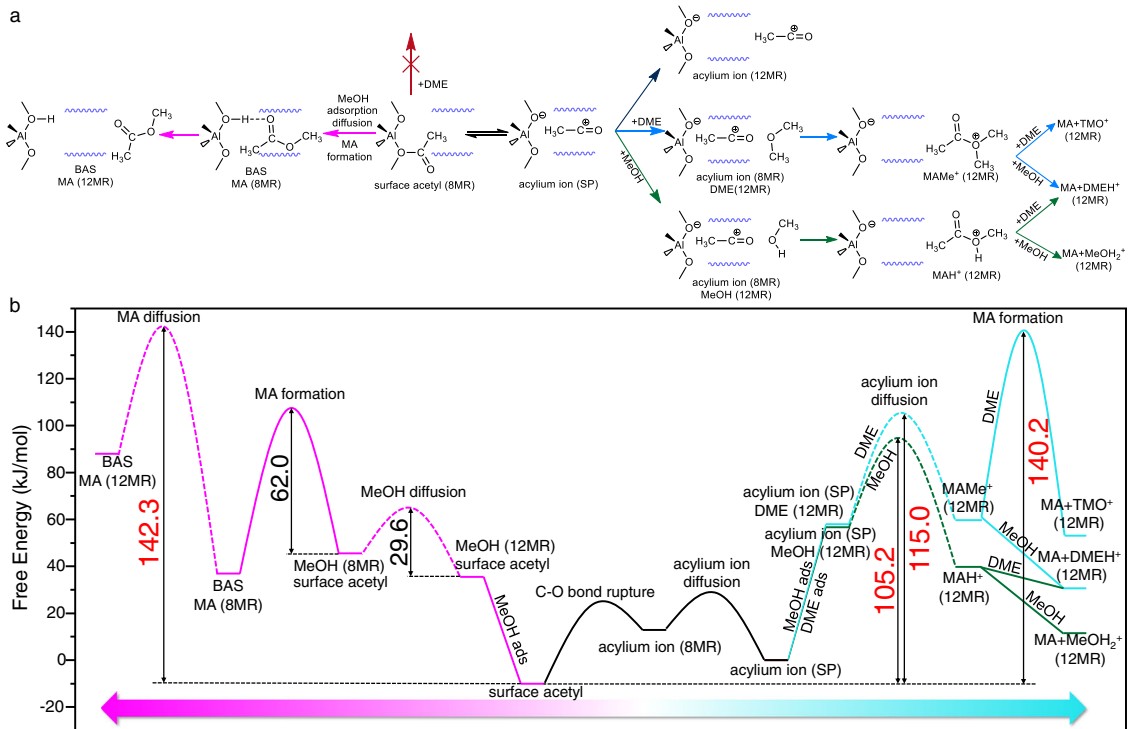

**Fig. 3 | MA formation based on the acylium ion in the side pocket (SP) and surface acetyl in the 8MR channel. a** reaction pathways, and **b** free energy profile. The dotted lines in **b** indicate the diffusion process; the values in black are the free energy barriers; the values in pink are the maximum free energy spans.

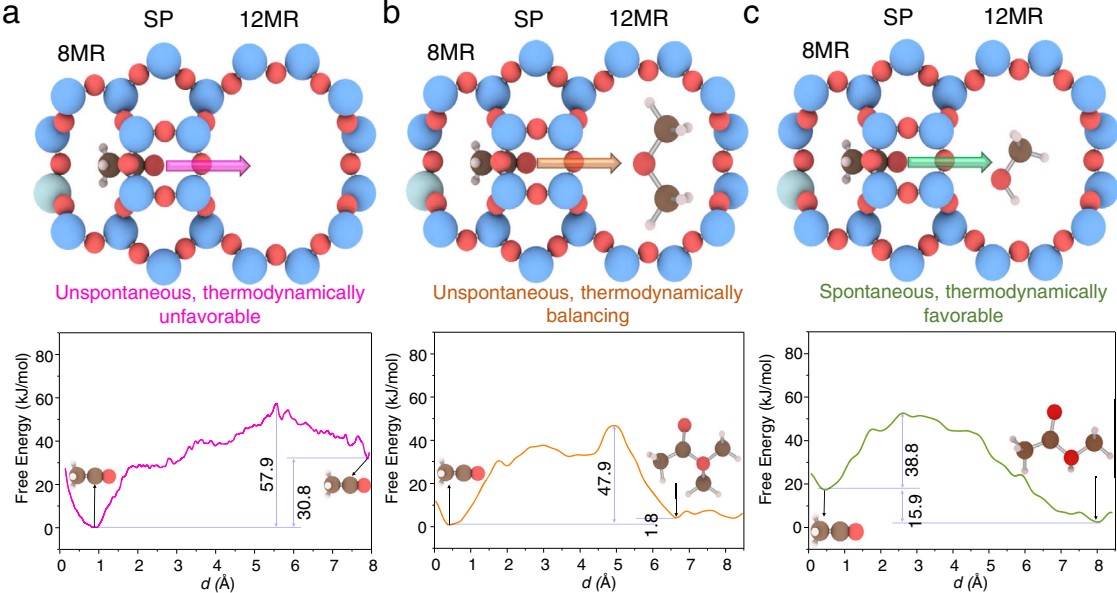

**Fig. 4 | The migration of the acylium ion from the side pocket to the 12MR channel. a** Free energy profile of the movement of the acylium ion from the side pocket to the 12MR channel without MeOH or DME in the 12MR channel, **b** free energy profile of the movement of the acylium ion into the 12MR channel previously occupied by DME and the further formation of MAMe⁺, **c** free energy profile of the movement of the acylium ion into the 12MR channel previously occupied by MeOH and the further formation of MAH⁺.

12MR channel for acylium ion migration results in the transfer of acylium ion to surface acetyl, however, the MA formation reaction will not occur spontaneously. (Supplementary Fig. 16, see more details of acylium ion diffusion in Supplementary Discussion), For that reason, Al atom in the 12MR channel of MOR will greatly affect the migration of acylium ion from 8MR channel to 12MR channel, and its elimination from the 12MR channel of MOR not only helps to avoid the C-C bond

coupling in MOR-12MR but also is be beneficial to the MA formation. In the successive reaction step, the charge-separated acylium ion in the 12MR channel would spontaneously couple with either DME or MeOH, thereby forming $CH_3COOCH_3R_1^+$ ($MAR_1^+$, $R_1 = CH_3$ or $H$), i.e., Eq. (2).

$$CH_3CO^+ + CH_3OR_1 \rightarrow MAR_1^+ \qquad (2)$$

$$MAR_1^+ + CH_3OR_2 \rightarrow MA + CH_3OR_1R_2^+ \qquad (3)$$

$$CH_3OR_1R_2^+ + Si-O-Al^- \rightarrow Si-OCH_3-Al + R_1OR_2 \qquad (4)$$

$$R_1 \text{ and } R_2 = H \text{ or } CH_3.$$

Once $MAR_1^+$ as the charged precursor of MA is formed in the 12MR channel, demethylation or dehydrogenation to MA can proceed via reaction with additional $CH_3OR_2$ (Eq. (3)). Note that these steps occur as barrierless and spontaneous processes during the regular AIMD simulations when $R_1$ or $R_2$ is H (Supplementary Fig. 17a–c). More interestingly, the DME unit in $MAMe^+$ underwent immediate substitution by MeOH, with the reaction of $MAMe^+$ and MeOH forming $MAH^+$ and DME products at ca 1.8 ps due to the weaker ester C-O bond in $MAMe^+$ than that in $MAH^+$. The deprotonation of $MAH^+$ occurs quickly since the ester oxygen is less basic than the oxygen of either DME or MeOH. However, the demethylation of $MAMe^+$ by DME is not a spontaneous process since methyl group transfer would require the inversion of the configuration in the methyl group of $MAMe^+$ to take place, which is an energetically demanding process. However, the cationic character of $MAMe^+$ reduces the $\Delta F$ to 69.9 kJ/mol with exothermal free energy of 6.9 kJ/mol (Supplementary Figs. 17d, 18). Overall, the formation of MA from $MAR^+$ and DME/MeOH is thermodynamically and kinetically favorable.

In addition, SMS species in MOR-8MR are more easily reformed due to the long-distance charge separation between $AlO_4^-$ in MOR-8MR and $CH_3OR_1R_2^+$ (~8 Å between two geometric centers). As shown in Supplementary Fig. 19, $CH_3OH_2^+$ (Eq. (3)) spontaneously diffuses from the 12MR channel to the 8MR channel via the side pocket under the influence of the long-range electrostatic interactions, while a low $\Delta F$ of 85.7 kJ/mol needs to be overcome to undertake the subsequent dehydration to SMS (Eq. (4)). Such a low $\Delta F$ is far smaller than that observed in traditional pathways (ca. 120 kJ/mol)[37,38].

Although MA formation in MOR-12MR can take place via surface acetyl and MeOH with low energy barriers, the barrier-less reaction pathway via the charge-separated acylium ion is still more energetically favorable. This is summarized in Fig. 5, which displays our proposed catalytic cycle of DME carbonylation in mordenite. In this new catalytic cycle, the various stages of the carbonylation in each cavity of mordenite act synergistically to form the product in a highly efficient manner. Of key importance here is the charge separation between $CH_3CO^+$ and $Si-O^--Al$ in the MOR-8MR, which effectively guarantees orderly MA formation, thereby avoiding aggregation of active intermediates, in contrast to the formation of ketene in MOR-12MR[39,40].

To consider the reasonability of newly proposed mechanism of MA formation, only the $\delta E$ of MA formation (105.2 kJ/mol and 115.0 kJ/mol) via migrated acylium ion and 2MeOH (or 1MeOH + 1DME) is energetically more favorable than other routes as listed in Table 1 when compared to the $\delta E$ of C−C bond coupling (128.6 kJ/mol) in MOR-8MR as the reference. Nevertheless, MA formation in MOR-8MR via MeOH and surface acetyl (142.3 kJ/mol) is still a more energy intensive process than C−C bond coupling. Therefore, we proposed that MA formation via acylium ion migration from 8MR to 12MR channel is a plausible mechanism that is more energetically favorable than the commonly accepted pathway despite MeOH or DME acting as the reactants. Notably, the high DME concentration (2DME) would also result in a large $\delta E$ (140.2 kJ/mol) of MA formation even using acylium ion as the intermediate, but the pathway of MA formation via acylium ion migration is thermodynamically more favorable than that via surface acetyl as displayed in Fig. 3b. Overall, DME is not a beneficial reactant, these results not only describe the priority of MA formation

via the migrated acylium ion but also emphasize the influence of different reactants to MA formation, and the more important role of DME as a reactant in carbonylation is highly possible to generate SMS and MeOH to facilitate the C-C bond coupling in MOR-8MR and MA formation in MOR-12MR, respectively.

## Solid-state NMR studies of acetyl species in mordenite

To confirm the above-mentioned mechanism, the solid-state NMR (ssNMR) experiments of acetyl chloride ($CH_3^{13}COCl$) adsorption on mordenite zeolite were employed to understand the interconversion of surface acetyl and acylium ion in mordenite zeolite as displayed in Fig. 6a. The acetyl chloride will rapidly transform to surface acetyl with the dechlorination on BAS in MOR-12MR ($CH_3^{13}COCl + Zeo-OH \rightarrow Zeo-O^{13}COCH_3 + HCl$) at 298 K when the partial pressure of $CH_3^{13}COCl$ is 0.5 kPa, and the signal of 183 ppm and 187 ppm can be assigned to the surface acetyl in 12MR channel and 8MR channel based on the reported 185 ppm by Li et al.[41]. The increasing concentration of acetyl chloride will further lead to the appearance of 167, 172, and 177 ppm, and 172 and 177 ppm can be assigned to unreacted acetyl chloride with the physical adsorption and hydrogen bond adsorption in 12MR channel based on the relationship of acetyl chloride concentration and signal strength[42], but the signal of 168 ppm can be attributed to acylium ion in side pocket based on our calculated value of 167 ppm. Due to the lower thermodynamical stability of acylium ion than surface acetyl (Fig. 3b), the signal strength of acylium ion is much lower than surface acetyl and acylium ion was formed after the formation of surface acetyl.

It is worth noting that the acetyl chloride cannot directly react with the BAS in 8MR channel of mordenite zeolite due to the size exclusion, but the appearance of 187 ppm (surface acetyl in 8MR channel) indicates the existence of a transferred mediator between 8MR and 12MR channels, i.e., linear ketene ($CH_2CO$). Herein, acetyl chloride was firstly dechlorinated to surface acetyl in 12MR channel, and then surface acetyl was deprotonated to ketene with high mobility and further diffused into the 8MR channel, and finally, ketene was protonated to surface acetyl in 8MR channel or acylium ion in side pocket by BAS in 8MR channel (Fig. 6c). To confirm this process, we further employed ssNMR to explore the evolution of acetyl species as displayed in Fig. 6b. The strength of 183 ppm (surface acetyl in 12MR channel) is weakened with the increasing time of heat treatment at 333 K but the signals of surface acetyl in 8MR channel (187 ppm) and acylium ion in side pocket (168 ppm) are enhanced. All these results confirm both our hypothesis illustrated in Fig. 6c and the relative thermodynamic stability and dynamic interconversion between acylium ion and surface acetyl in MOR-8MR (Fig. 3b).

## in situ FTIR studies of carbonylation in mordenite

Under the premise of the theoretical results and ssNMR experiments, Fourier transform infrared (FTIR) spectroscopy was employed to confirm this new charge-separation driven mechanism in the carbonylation of MeOH and CO in mordenite (Supplementary Fig. 20). The interaction of MeOH with BAS via hydrogen bonding at 363 K led to the consumption of BAS in the 12MR and 8MR channels as evidenced by the erosion of the bands at 3610 and 3580 $cm^{-1}$, respectively (see negative bands in Fig. 7a). The subsequent admittance of CO molecules over H-mordenite with pre-adsorbed methanol resulted in the appearance of the 2220 $cm^{-1}$ band of $C\equiv O$, which is slightly covered by the rotation-vibration spectrum of gaseous CO (Fig. 7c). The reaction between MeOH and CO is also documented by the appearance of two new C–H stretching ($\nu_s(C\text{-}H)$) vibration peaks at 2974 and 2867 $cm^{-1}$ in addition to those already originating from pre-adsorbed MeOH (or SMS) and present at 2954 and 2855 $cm^{-1}$ (Fig. 7b). The newly formed C−H bands (2974 and 2867 $cm^{-1}$) together with the $\nu_s(C\equiv O)$ band (2220 $cm^{-1}$) were assigned to the acylium ion based on our previous acetyl chloride (AcCl) adsorption experiments that located the acylium

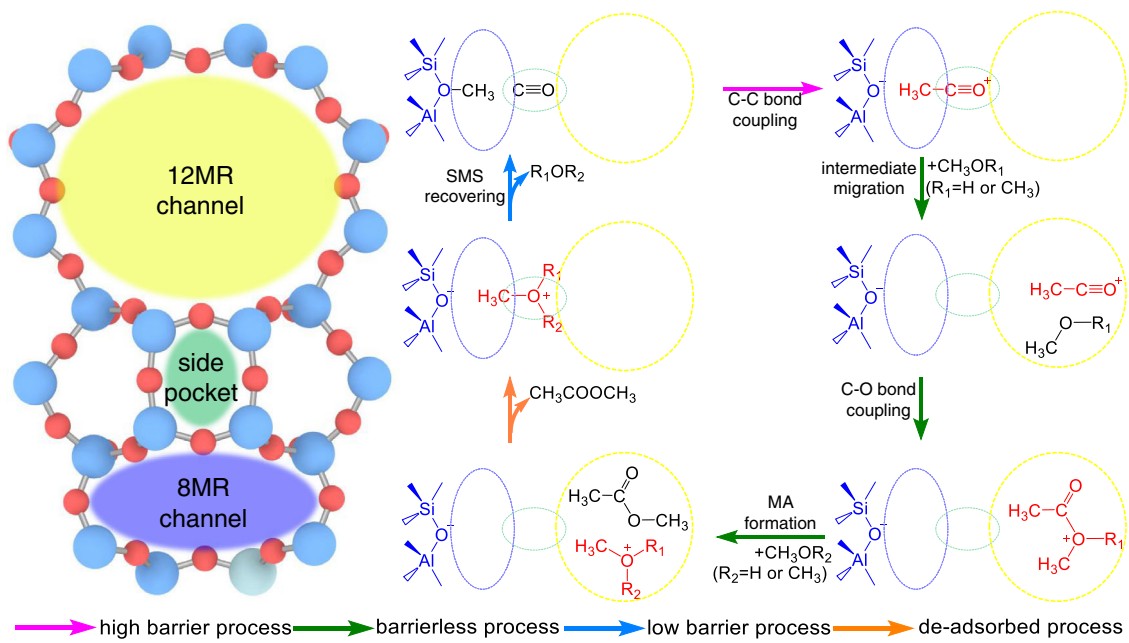

**Fig. 5 | Mechanism of DME/MeOH carbonylation through the synergistic action of 8MR channels, side pocket, and 12MR channels in mordenite zeolite.** Positively and negatively charged chemical species are depicted in red and blue, respectively.

ion band at 2285 cm$^{-1}$[27]. The lower wavenumber observed in the current case is explained by the HCl and unreacted AcCl as co-adsorbed species. After prolonged time (30 min at 363 K), the acylium ion was totally transformed to surface acetyl in 12MR ($v_s$(C–H): 2974 and 2867 cm$^{-1}$; $v_s$(C = O): 1630 cm$^{-1}$) and 8MR ($v_s$(C–H): 2964 cm$^{-1}$; $v_s$(C = O): 1605 cm$^{-1}$) and MA/AcOH ($v_s$(C–H): 3250 cm$^{-1}$, $v_s$(C = O): 1670 cm$^{-1}$). The factors indicate that acylium ion in the side pocket is rapidly converted into surface acetyl or directly produce MA/AcOH. As a result, MA/AcOH is formed via both acylium ions and surface acetyl species, following the above outlined simulated results.

Notably, C–C bond coupling between SMS and CO is energetically more favorable in MOR-8MR than in MOR-12MR. Indeed, the acylium ion is the initial intermediate of C–C bond coupling in MOR-8MR as documented by the appearance of the band at 2220 cm$^{-1}$ immediately after addition of CO at 363 K ((iii) in Fig. 7c), while increasing the temperature further to 473 K led to formation of surface acetyl species ((iv) in Fig. 7d). These results are highly consistent with their relative thermodynamic stabilities again (Fig. 3). As the acylium ion is an intermediate product with a higher activity than surface acetyl, it can easily form MA or AcOH at already low temperatures. This is documented by the FT IR spectra showing that the acylium ion in the side pocket disappears at higher temperatures due to conversion into more stable surface acetyl, with concomitant formation of MA or AcOH through reaction with MeOH. Our studies prove the role of the stable

surface acetyl in the formation of MA and AcOH in mordenite, while the acylium ion intermediates are reported as particularly high-active species in the formation of MA and AcOH in mordenite.

## 2-D COS rapid scan FTIR studies of carbonylation in mordenite

Further confirmation of the temporal and spatial correlations between the different species involved in MeOH carbonylation was achieved from 2D COS analysis of the FT-IR spectra collected during the transformations of stoichiometric mixture of CO and MeOH in mordenite at 493 K. Two-dimensional correlation (2D COS) analysis is based on the spreading of the spectrum over the second frequency which increases resolution of the overlapping peaks, further facilitating and enhancing their detection and characterization[43,44]. Herein, the role of the acidic sites (BAS and silanols) during the carbonylation process and the evolution of the surface species in different channels were studied in detail (Fig. 8 and Supplementary Fig. 21).

First, we observed that the carbonylation process leads to the simultaneous consumption of silanols (3745 cm$^{-1}$) together with BAS located in 12MR (3615 cm$^{-1}$) (Supplementary Fig. 21a). Secondly, a negative correlation between the BAS bands in 12MR and side pockets (3615 cm$^{-1}$ and 3595 cm$^{-1}$, respectively) and 3585 cm$^{-1}$ in 8MR (Fig. 8a) indicates for a synchronous consumption of BAS in 12MR and BAS restoration in 8MR. Thus, the reaction intermediates are released from the BAS located in 8MR while their abundance increases in 12MR, directly revealing the different roles of 8MR and 12MR channels of mordenite in carbonylation.

The simultaneous involvement of the BAS both in 8MR (3585 cm$^{-1}$) and side pocket (3595 cm$^{-1}$) in the formation of acetyl species is evidenced by the negative correlation between the above-mentioned O–H vibration bands and the C–H vibration bands in the acetyl group (2974 and 2865 cm$^{-1}$) (Fig. 8b, c). Here, the band at 2865 cm$^{-1}$ was attributed to acetyl species in 8MR channels. The SMS are consumed in accompany with BAS in 8MR, as manifested by the positive correlation between the SMS (2855 cm$^{-1}$) and the BAS sites (3585 cm$^{-1}$) in 8MR. In turn, the disappearance of SMS in 12MR (2855 cm$^{-1}$) is followed by the restoration of BAS in the same channels (3615 cm$^{-1}$). The positive correlation between the bands of to acetyl (2974 and 2865 cm$^{-1}$) and SMS (2855 and 2954 cm$^{-1}$) species in the 2-D COS map of the 3100–2700 cm$^{-1}$ × 3100–2700 cm$^{-1}$ regions (Supplementary Fig. 21d)

**Table 1 | Maximum free energy span (kJ/mol) of MA formation via surface acetyl and DME/MeOH in mordenite at 473 K, as well as acylium ion and DME/MeOH in MOR-8MR**

| Reactants | C–C bond coupling[a] SMS CO | MA formation (this work) | | | | |
|---|---|---|---|---|---|---|
| | | surface acetyl | | acylium ion | | |
| | | MeOH | DME | 2MeOH | 2DME | 1MeOH + 1DME |
| 8MR | 128.6 | 142.3 | >249.6[b] | 105.2 | 140.2 | 105.2 or 115.0[c] |
| 12MR | 155.7 | 138.7 | 165.5 | | | |

The adsorption free energies were considered.
[a]Data from ref. 27.
[b]Data from the static DFT calculation in this work.
[c]Firstly MeOH and then DME, and vice versa.

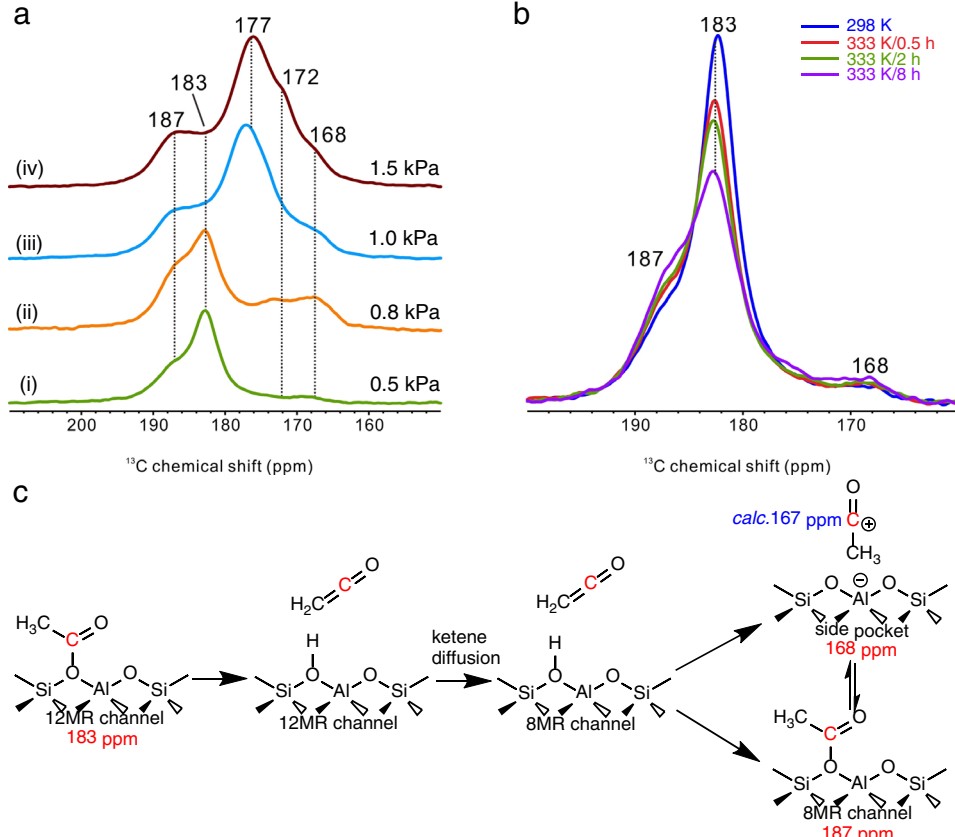

**Fig. 6 | ssNMR spectra of CH₃¹³COCl reacted with dehydrated H-MOR zeolite and the proposed scheme of the intermediates evolution. a** ¹³C MAS NMR spectra of different loadings at 333 K for 2 h. (i) 0.5 kPa, (ii) 0.8 kPa, (iii) 1.0 kPa, and (iv) 1.5 kPa. **b** In situ ¹³C MAS NMR spectra of CH₃¹³COCl adsorbed on dehydrated H-MOR zeolite at different reaction conditions. **c** Assignation to chemical shifts, and the evolution of ketene, surface acetyl and acylium ion in different channels.

confirms that they both undergo simultaneous reactions. The transformation of SMS in acetyl species is rapid enough to be beyond rapid scan FTIR detection level.

The presence of surface acetyl is also accompanied by MA or AcOH, as confirmed by the presence of the MA and AcOH bands in the deformation mode range ($v_d$(MA or AcOH), 1800–1300 cm⁻¹) (Fig. 8e, f, and Supplementary Fig. S21e). Accordingly, the bands at 1437 and 1465 cm⁻¹ are assigned to bending vibrations of the CH₃ groups ($v_b$(−CH₃)) in surface acetyl, while the 1630 and 1608 cm⁻¹ bands are ascribed to the $v_s$(C=O) of surface acetyl located in MOR-12MR and MOR-8MR, respectively. Thus, it is observed that changes in the chemical nature of the final reaction products determine the nature of the C−H stretching vibrations. The final products accumulation on the catalyst surface is documented by the positive correlations solely. This buildup of reaction products is related to the consumption of silanols and BAS in the 12MR channel only (the negative correlations in Fig. 8d, g). In turn, the reaction products formation is related with the BAS in both the 8MR channel and the side pockets occupation (the positive correlations in Fig. 8d).

As a result, the accumulation of MA in MOR-12MR and on the outer surface of the zeolite crystal is in conformity with our proposed mechanism as outlined in Fig. 5. Consequently, the sites that most selectively participate in the transformation of surface acetyl to the final carbonylation products are BAS and surface acetyl in MOR-8MR.

## Discussion
Through AIMD simulations using enhanced sampling methods, we comprehensively investigated MA formation via different intermediates (surface acetyl and acylium ion) in MeOH/DME carbonylation

over mordenite. Due to the high activity and mobility of the acylium ion in the side pocket of mordenite, MA formation via the acylium ion is both thermodynamically and kinetically more favorable than the traditional pathway via surface acetyl. In the newly proposed pathway of MA formation, the acylium ion in the side pocket can easily migrate into MOR-12MR and spontaneously form MA. Notably, this proposed mechanistic process is energetically more favorable than MA formation without intermediate migration (i.e., C−O bond coupling between MeOH/DME and surface acetyl). Confirmation of this mechanism was first achieved by the observation of dynamic evolution between surface acetyl and acylium ion in 8MR channel in ssNMR experiment. More importantly, direct spectroscopic evidence for the key role of the 8MR channel in the formation of the reaction intermediates and the migration of reactive species to the 12MR channel was obtained through 2D COS FTIR analysis. The distinguishing feature of the new mechanism is the presence of ionically charged acylium ions and deprotonated BAS; nevertheless, the strong electrostatic interaction between these two species is largely mitigated by the confinement effect of the side pocket. It is precisely the presence of these ionically charged species that is responsible for the highly efficient catalytic cycle and the long catalyst life of mordenite during carbonylation. In this context, each different cavity and channel in mordenite has a very specific role in the catalytic cycle of carbonylation. The BAS in the 8MR channel efficiently promotes acylium ion formation via C-C bond coupling between the SMS and CO. The side pocket plays the important role of stabilizing the acylium ion while driving its mobility due to steric constraints towards the 12MR channel without much compensatory energy. Finally, the 12MR channel is sufficiently large for MA formation via C−O bond coupling between the acylium ion and MeOH/DME.

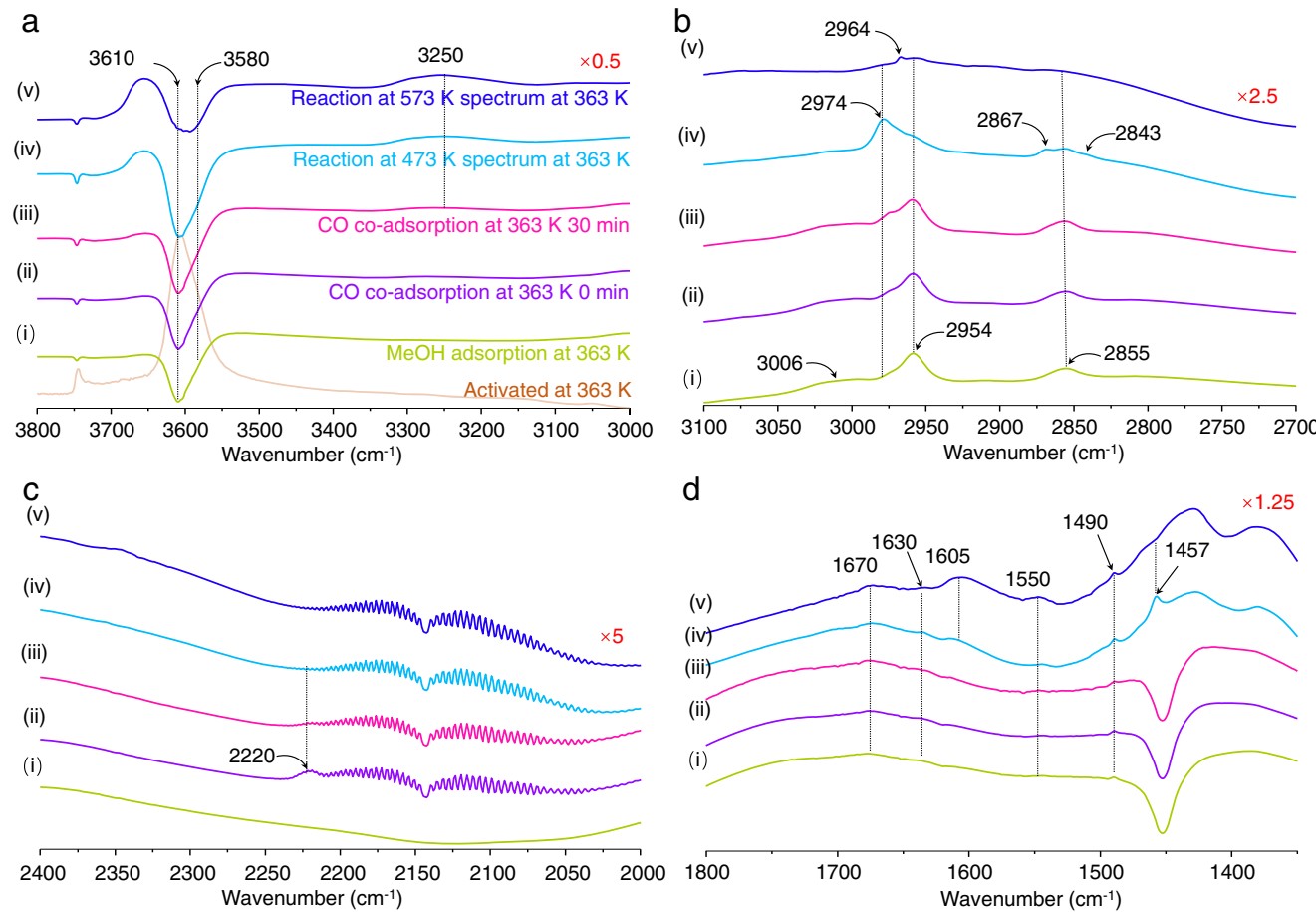

**Fig. 7 | FTIR spectra of carbonylation substrates adsorbed in mordenite.** The different collected conditions of FTIR spectra: (i) MeOH adsorption at 363 K, (ii) 0 min and (iii) 30 min co-adsorption MeOH and CO at 363 K, (iv) carbonylation reactions at 473 K and (v) carbonylation reactions at 573 K. **a** O–H stretching vibrations: 3800–3000 cm⁻¹, **b** C–H stretching vibrations: 3100–2700 cm⁻¹, **c** C≡O stretching vibrations: 2300–2000 cm⁻¹, and **d** carbonyl frequencies: 1800–1350 cm⁻¹.

Our newly proposed mechanism acts through three pore architectures acting synergistically in differently confined environments to promote the efficient carbonylation in mordenite via the mobile acylium ion intermediate. Notably, the proposed intermediate migration mechanism can also be generalized to other zeolite catalyzed reactions. Analogously to acylium ion, the carbocation as one kind of common intermediate in zeolite catalyzed reactions (like methanol to olefins, alkene cracking, alkylation, etc.) also has strong mobility, especially at high reaction temperature, and the mobilizable carbocation with the charge separation between carbocation and $AlO_4^-$ lead to the high reactivity of carbocation kinetically, and the confined environment of zeolite framework can help to stabilize carbocation thermodynamically[45]. In addition, this new mechanism was proposed by an integrated reaction/diffusion kinetics model, and all processes (adsorption and desorption, reaction, and diffusion) of the complete catalytic cycle were involved in a free energy landscape, which makes the integrated consideration to reaction mechanism possible. Such model can also be applied to understand the product selectivity of zeolites in the reaction of MTO, syngas conversion, $CO_2$ reduction, etc.

## Methods
### Theoretical calculations
The topologies of mordenite were obtained from the International Zeolite Association website. In the mordenite cell ($a = 18.26$ Å, $b = 20.53$ Å, and $c = 15.08$ Å), one of the silicon atoms in T4-O10-T4 (12MR) or T3-O8-T3 (8MR) was replaced by aluminum to introduce the BAS in the 12MR and 8MR channels, respectively. Correspondingly, the final materials were referred to as MOR-12MR and MOR-8MR. To fully

account for the flexibility of the zeolite framework and the dynamics of the guest molecules under the reaction temperature, AIMD simulations were performed using the DFT. The temperature of the AIMD was controlled using a chain of five Nosé-Hoover thermostats[46], and the integration time step was set to 0.5 fs. Two enhanced sampling methods, metadynamics (MTD) and umbrella sampling (US), were employed to simulate the reaction and mobility, respectively. All AIMD simulations were conducted using the CP2K software of version 5.1[47], and the linked PLUMED code of version 2.7 was used to carry out the MTD and US methods[48]. The Perdew–Burke–Ernzerhof (PBE) functional[49] with consideration of Grimme D3 dispersion corrections[50], i.e., the PBE-D3 functional, was chosen for the DFT calculations. Limited by large computational cost of triple-zeta basis set in AIMD simulations, the double-zeta ($\zeta$) valence polarized (DZVP) basis set[51] together with the Goedecker–Teter–Hutter (GTH) pseudopotential[52] were used for the system. During the self-consistent field (SCF) procedure, a 360 Ry density CUTOFF criterion with the finest grid level was employed along with multi-grids number 4 (NGRID 4 and REL CUTOFF 70), and the SCF convergence criterion was set to $10^{-9}$ a.u.

Before the calculations of the reaction process, a 5 *ps* equilibrium AIMD simulation of the reactant state was carried out under the isothermal-isobaric (NPT) ensemble to ensure the relaxation of the cell parameters and the atom positions. It should be noted that reactions taking place in this equilibrium stage were considered as spontaneous reactions (barrier-less process) under the reaction conditions. However, the non-spontaneous reaction was explored by MTD method employed for enhancing the probability of sampling chemical reactions or rare events provided that a limited number of collective

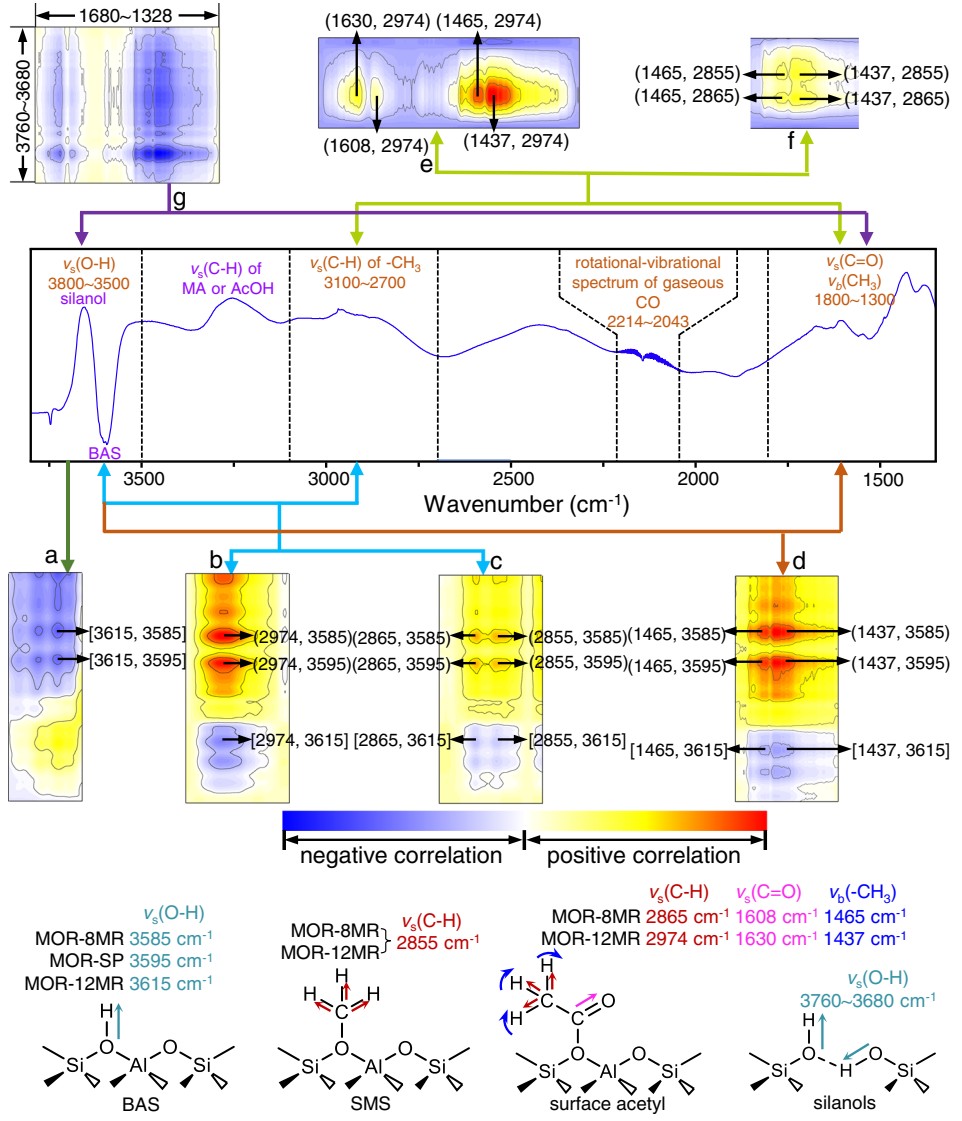

**Fig. 8 | 2D COS maps of the rapid scan FT-IR spectra.** Positive and negative correlations between $v_1$ and $v_2$: **a** $v_s$(O–H) of BASs in MOR-12MR and $v_s$(O–H) of BAS in MOR-8MR and MOR-SP, **b** $v_s$(C–H) of surface acetyl in MOR-12MR and $v_s$(O–H) of BASs, **c** $v_s$(C–H) of surface acetyl in MOR-8MR and $v_s$(O–H) of BASs, **d** $v_b$(–CH$_3$) of surface acetyl and $v_s$(O–H) of BASs, **e** $v_s$(C–H), $v_s$(C = O), and $v_b$(–CH$_3$) of surface acetyl in MOR-12MR, **f** $v_s$(C–H) of surface acetyl and SMS in MOR-8MR and $v_b$(–CH$_3$) of surface acetyl, **g** $v_s$(O–H) of silanols and $v_d$(MA or AcOH). Coordinates in square brackets ([#]) are negative correlations, while those in standard brackets ((#)) are positive correlations.

variables (CV) describing the reaction coordinates were well-defined as illustrated in Supplementary Fig. 4. Nonetheless, MTD simulation is normally biased by regularly spawning Gaussian hills along the chosen CVs, which was defined by coordination numbers (CN):

$$\mathrm{CN} = \sum_{ij} \frac{1 - \left(\frac{r_{ij} - d_0}{r_0}\right)^n}{1 - \left(\frac{r_{ij} - d_0}{r_0}\right)^m} \tag{5}$$

where $r_{ij}$ represents the interatomic distance between atoms $i$ and $j$ and $r_0$ is the reference distance. In this work, parameters $d_0$, $n$, and $m$ were set to 0, 6, and 12, respectively. The 2D free energy profile was reproduced according to the bias potential of MTD algorithm implemented within PLUMED, that is defined as:

$$F(S) = -k_B T \left( \int dR \, \delta(S - S(R)) e^{-\frac{U(R)}{k_B}} \right) \tag{6}$$

$k_B$ is the Boltzmann constant, $T$ is the temperature of the system, $U(R)$ is the potential energy function, and S(R) is the CVs.

Gaussian hill width and height of all MTD simulations were summarized in Supplementary Table 2, MTD simulations were allowed to continue till the height of additional hills no longer affect the resultant free energy profile, and the MTD simulations were considered to be converged if the barriers between every 500 hills added do not differ by more than 1 kJ/mol as displayed in Supplementary Fig. 22. At least 50 ps AIMD-MTD simulations were used to fully guarantee the reaction equilibrium. Based on the sum of spawned Gaussian hills, the 2D free energy profile of the reaction may be reconstructed. Accordingly, the lowest free energy path (LFEP) was derived by means of the MEPSA software[53].

In the US, the mobilities of the reactant species from the 8MR channel to the 12MR channel were defined by a collective variable, ((the distance between the two centers of mass (Supplementary Figs. 5–7)), that is, the reactant species and the 8MR window. The additional

potential introduced by a simple harmonic term:

$$U(s) = \frac{k}{2}(x - x_0)^2 \qquad (7)$$

where $x$ is the real-time value of CV, $x_0$ is the value of specified value of CV, $k$ is set to 50 kJ/mol for acylium ion, DME, MeOH, and MA. All biased umbrella sampling simulations were consisted $10^4$ steps at NVT ensemble, and the setting to $k$ lead to the reasonable sampling indicated by the inter-overlapping of different windows. The free energy profile was generated by the sampling obtained in each window based on the weighted histogram analysis method[54]. The error bars of free energy profiles generated from MTD and US simulations have been calculated as displayed in Supplementary Figs. 23, 24, and more details of sampling analysis were illustrated in Supplementary Methods and Supplementary Figs. 25, 26.

### Solid-state NMR experiments

The H-form mordenite (H-MOR) zeolite sample used herein was obtained from the parent $NH_4$-MOR zeolite (Zeolyst, CBV-21A; Si/Al = 10). To obtain H-form H-MOR zeolite, the $NH_4$-MOR sample was carefully deaminated by gradual heating of the tube furnace in dry air at a temperature rate of 1 K/min. The sample was firstly dried at 383 K for 2 h, then calcinated at 773 K for 3.5 h.

Prior to the adsorption of $CH_3^{13}COCl$, each zeolite sample was placed in a home-made glass tube and then connected to a vacuum line for dehydration treatment. Typically, this was performed by gradually increasing the temperature from room temperature to the final target temperature (673 K) at a rate of 1 K/min. The sample was then kept at 673 K under a pressure of $<10^{-3}$ Pa for at least 10 h and then cooled. After the sample was finally cooled to ambient temperature, a known amount of volatile $CH_3^{13}COCl$ molecule was introduced into the activated sample in a liquid $N_2$ bath and the sample tube was then flame-sealed. The reaction of different loadings of $CH_3^{13}COCl$ with dehydrated H-MOR zeolite was performed by further baking the sample tube at 333 K for 2 h. Prior to NMR experiments, the sample in the sealed glass tube was transferred into a $ZrO_2$ rotor under a dry nitrogen atmosphere in a glovebox and then sealed with a Kel-F end-cap. It should be noted that the in situ experiment of 0.5 kPa $CH_3^{13}COCl$ with dehydrated H-MOR zeolite was achieved by directly baking the sealed $ZrO_2$ rotor in a 333 K oven.

All the solid-state $^{13}C$ NMR experiments were performed on a Bruker AVANCE 800 MHz spectrometer operating at a Larmor frequency of 800.36 and 201.27 MHz for the $^1H$ and $^{13}C$ nucleus, respectively, with a 3.2 mm magic-angle-spinning (MAS) probe at a spinning rate of 16 kHz. $^{13}C$ MAS NMR spectra with high power proton decoupling were recorded with a π/2 pulse length of 3.3 μs, a recycle delay of 5 s, and 512 scans. The chemical shift of $^{13}C$ nucleus was externally referenced to adamantane.

### FTIR experiments

Prior to proceeding with the carbonylation experiment via adsorption of MeOH and CO, H-MOR (Si/Al=10) was shaped into a thin wafer (~8.5 mg/cm²) and placed in a custom-made quartz IR cell connected to a high vacuum line for evacuation purposes. Thermal activation of the sample was carried out by gradually increasing the temperature at a rate of 1 K/min, with a final temperature of 673 K and a pressure of $<10^{-5}$ Pa for 2 h. After the sample was cooled down to 363 K, specific doses of methanol were introduced into the sample to assure the half saturation of the BAS. Next, the physiosorbed methanol molecules were removed via evacuation at the same temperature. The methanol pretreated zeolite was exposed to a stoichiometric amount of CO for 90 min. Then, the temperature was increased to 473 K. The FTIR spectra were acquired in rapid-scan mode (80 kHz) using a Vertex 70 spectrometer (Bruker). Each spectrum (5 scans) was collected within 1 sec, with a spectral resolution of 2 cm⁻¹.

### 2D COS analysis of FTIR spectra

The reaction between $CH_3OH$ and CO on the H-MOR catalyst was followed by in situ studies using rapid scan FTIR spectroscopy. The catalysts were thermally activated as previously described, and stoichiometric amounts (1:1) of the substrates were immediately introduced in the spectroscopic quartz cell at a temperature of 497 K and an absolute pressure of 520 Pa. The rapid scan FTIR spectra were recorded every 0.5 s with a resolution of 2 cm⁻¹ during a reaction period of 10 min. Furthermore, 2D COS analysis in synchronous mode was performed on selected regions of the spectra. The 2D COS analysis was performed on the differential FTIR spectra obtained after the introduction of the reagent and the spectra of the activated sample was subtracted.

## Data availability

The data that support the findings of this study are available from the corresponding authors on request.

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

## Acknowledgements

This research was supported by the National Key Research and Development Program of China 2021YFA15026009 (X.Y.); the National Natural Science Foundation of China 22125304 (A.Z.), 22032005 (A.Z.),

22002174 (W.C.), 21991092 (A.Z.), 21902180 (Z.L.), 22202215 (X.T.); the National Science Centre, Poland, Grant No 2021/41/B/ST4/00048 (K.G.M.) and Grant No 2020/37/B/ST4/01215 (K.A.T.). The AIMD simulations in this work was carried out at TianHe-1(A) of National Supercomputer Center in Tianjin, and the EDA calculations in this work were supported by the supercomputers in Shantou University.

## Author contributions

W.C., K.G.M., and A.Z. contribute conceived the project, W.C. designed all theoretical simulations, X.Y. carried out the solid-state NMR experiment; K.A.T. and K.G.M. performed the infrared spectroscopy experiments; Z.L. and X.T. contributed the important discussions and suggestions to the simulated results; W.C., K.G.M., and A.Z. wrote the manuscript.

## Competing interests

The authors declare no competing interests.
