## [Peer Review File · Nature Communications]

Charge-separation Driven Mechanism via Acylium ion
Intermediate Migration during Catalytic Carbonylation in
Mordenite ZeoliteReviewers' comments:

Reviewer #1 (Remarks to the Author):

Ms. Ref. No.: NCOMMS-22-21836

Title: Charge-separation Driven Mechanism via Acylium ion Intermediate Migration during Catalytic Carbonylation in Mordenite Zeolite

Authors: Wei Chen, Karolina A. Tarach, Xianfeng Yi, Zhiqiang Liu, Xiaomin Tang, Kinga Góra-Marek, and Anmin Zheng

In the submitted work, the authors propose a new favorable pathway for the formation of methyl acetate due to the synergistic action of 8MR and 12MR of mordenite. The acylium ion created in the carbonylation of the surface methyl group diffuses to 12MR to undergo a further reaction with MeOH/DME. Products of this reaction are methyl acetate and protonated methanol (CH_3OH_2^+), which diffuses back to 8MR and restores the surface methyl group. This study therefore provides a novel and very interesting perspective on the research subject, as in previous studies the preferred mechanism was traditionally explained via less energetically efficient transformations involving surface acetyl in 8MR, while the formation of C-C bonds in 12MR was attributed only to the formation of coke.

The authors used advanced ab initio molecular dynamics simulations to explore the reactions and effects involved in this novel pathway; however, I have some points that should be addressed by the authors before I can recommend the manuscript for publication:

1) For simulations in 12MR and 8MR, the authors used two different zeolite models with different positions of Al atoms in the framework. The presence or absence of Al in channels and cavities affects the energetics of reactions and processes that occur in the catalyst. Therefore, a direct comparison of processes such as diffusion, which was studied in the submitted work, might be affected by this.

The model having Al in 8MR has different properties than the model with Al in 12MR, and also a model with two Al atoms in both 8MR and 12MR, which was not considered in this study, but in which it could be assumed that the additional Al atom would be able to stabilize some intermediates. Could the authors comment on to what extent the position of the Al atom affects the results? In addition, could the authors provide more detailed figures of cells with highlighted positions as a part of SI.

2) In the submitted work, as shown in Scheme S1, the authors used a different CV for MOR-8MR in the MTD simulation than for MOR-12MR and SSZ-13 in the case of MA formation via surface acetyl. The choice of a good CV that properly describes the progress of reactions can be a very difficult task. In the work of B. Peters (<https://doi.org/10.1146/annurev-physchem-040215-112215>), the properties and criteria which a good reaction coordinate should fulfill are discussed. It would be useful if the authors commented on their choice of CVs that were used in their work and how they confirmed their quality.

3) The authors state that the formation of surface acetyl in 8MR is hindered and energetically unfavorable, however, in the work of M. Gešvandtnerová et al. (<https://doi.org/10.1016/j.jcat.2021.02.011>) surface acetyl was discovered from molecular dynamics simulation of acylium ion after some time in the side pocket. This suggests that the formation of surface acetyl in 8MR is energetically feasible, and the barrier for the conversion can be overcome even without treating the transformation like a rare event with a need for specialized MD techniques. This makes me wonder how did the authors know when to stop the equilibration period of their simulations, and how did the authors make sure that their simulations are long enough?

4) As a part of the main text, the authors provide a section “Methods” with a subsection “Theoretical Calculations” in which the authors refer the reader to their previous work for computational details. This section should be extended, or can be added as a part of SI, with relevant computational settings and details, such as the particular DFT functional, convergence criteria and length of their simulations for MTD and for US windows.

5) The authors came to their conclusions by comparing the energy spans calculated from the free energies obtained from the free energy profiles for reactions consisting of multiple steps. Energy of each step is usually calculated with some error, which is cumulated in the resulting energy span. Thus, I am interested in what accuracy was achieved for the results and if the energy difference is still statistically significant even when errors are included. Could the authors comment on the accuracy of their energy spans and their estimation of errors?

The authors also consider the interconversion of surface acetyl to acylium ion, with surface acetyl being energetically lower (shown in Fig. 4). Calculating the energy spans of two competitive reactions is appropriate if they share the same most stable determining intermediate. Therefore, the energy span for the right side of the profile (Fig. 4) should be calculated with surface acetyl as a determining intermediate, as it is energetically lower, and therefore it maximizes the energy span, as described in the work of Kozuch et al. (<https://doi.org/10.1021/ar1000956>). Energy span for the right side of the profile in Fig. 4 calculated this way is comparable to the energy span for the left side of the profile. Taking into account the possible errors, these values could be equivalent, and then it is questionable if such a clear final conclusion can be made.

6) This point follows up the previous point concerned with the accuracy of the calculated free energies. From the computational details that are provided, it is not clear to me how the authors took into account the terms of the probability densities of their reactant configurations and velocities in the transition states. Here, I am referring to Eq. 7 in the work of M. Gešvandtnerová et al. (<https://doi.org/10.1016/j.jcat.2021.02.011>), and to the works of E.A. Carter et al. ([https://doi.org/10.1016/S0009-2614\(89\)87314-2](https://doi.org/10.1016/S0009-2614(89)87314-2)), and T. Bučko et al. (<https://doi.org/10.1039/C7CP05562E>).

Minor points:

7) Is there a reason why free energy profiles are called 'free energy surface' and vice versa? The profile in Fig. 1c is denoted in the caption as 'surface' and the free energy surfaces in Fig. 2 are denoted as 'free energy profiles'. This point is valid also for Figures 3, S1, S3, S5, S9, S10 and S11.

8) Movies are mentioned in the main text, but no movies were provided in the zip file for reviewers.

9) The color code in Fig. 4 is not properly chosen – the pink values are not pink, but purple, however the lines are pink. This is a bit chaotic.

10) The caption for Fig. 8 is missing (iii) – (v).

Reviewer #2 (Remarks to the Author):

This manuscript deals with investigations of the reaction mechanism of CO methylation in the zeolite Mordenite using mostly molecular dynamics simulations but also IR spectroscopy. Based on the MD data new pathways and intermediates are suggested for the formation of methyl acetate including previously identified molecules such as ketene.

Overall, I find this study very specific, even for zeolite catalyzed reactions, as this is investigating one key step for one specific zeolite morphology. I hence do not see how this manuscript should be of vivid interest to the broad audience of Nature Communications, nor do I see the "smoking gun" that makes the findings here a hot topic. I therefore recommend to reject this manuscript and suggest resubmission to a catalysis specialized journal, e.g. Journal of Catalysis.

Reviewer #3 (Remarks to the Author):

Wei et al. investigated methyl acetate (MA) formation via different intermediates in MeOH/DME carbonylation over mordenite and SSZ-13 through AIMD simulations using enhanced sampling methods. With the high activity and mobility of the acylium ion in the side pocket (SP) of mordenite, it is more thermodynamically and kinetically favorable for MA formation via acylium ion than the traditional pathway with surface acetyl. With employing advanced ab initio molecular dynamic simulations and two-dimensional correlation analysis of rapid scan infrared spectroscopy, this work has proposed a new pathway for MA formation via acylium ion in the 12MR channel. In this pathway, the acylium ion in SP can easily migrate into MOR-12MR and spontaneously form MA. This newly proposed mechanism acts through 3 pore architectures acting synergistically in differently confined environments to promote the efficient carbonylation in mordenite via mobile acylium ion intermediate. It can be confirmed with the direct spectroscopic evidence that the 8MR channel plays a key role in MA formation and the migration of reactive species to the 12MR channel can also be found through 2-D COS FTIP analysis. However, there are major concerns about the theoretical work for possible publication. The detailed comments are provided as following:

1. Fig.1d showed the energy decomposition analysis between organic species and zeolite framework. However, the authors didn't clearly explain how they defined and calculated E_{Pauli} , E_{elect} , E_{orb} , E_{disp} and E_{total} . More comprehensive explanations are needed here. In addition, the motivation for authors to compare these parameters are also needed here.
2. The author applied MA formation over SSZ-13 mechanism for comparison with that over MOR. But SSZ-13 and MOR have many different factors that could lead to the different catalytic performance of reactions, i.e., confinement, membrane size, transport limitation, acid/base site density. Please clarify.
3. More motivation details are needed for Fig. 2 to employ the coordination number for illustrating the reaction mechanism.
4. On Page 11, the authors said that the surface acetyl and acylium ions are the possible intermediates connecting C-C bond coupling and MA formation in MOR-8MR. However, the following part gave an assumption that surface acetyl is the sole reacting species. Please strengthen this assumption with experimental evidence. Also, why not take the acylium ions into consideration in this part?
5. The selectivity of zeolite materials is influenced by the channel sizes and molecule sizes. Seen in Fig. 7, a new mechanism for MA formation with acylium ions migration has been provided. Is that possible for acylium ions bonding with surface methyl species in 8MR or 12MR side rather than in SP?

Referee 1: additional AIMD simulations have been carried out to explore acylium ion migration in the presence of additional BAS.

Referee 2: the migrated charged intermediate mechanism has been expanded into a common mechanism in zeolite catalysis, and the used integrated reaction/diffusion kinetics model has been proposed as a new method to obtain the complete reaction mechanism of catalytic cycle.

Referee 3: NMR experiments have been replenished as strong evidence to support our mechanism in this work.

I hope that the changes we made are sufficient to make our paper acceptable for publication, and greatly appreciate your reconsideration.

With best wishes from Wuhan!

Anmin Zheng

Reviewers' comments:

Reviewer #1 (Remarks to the Author):

In the submitted work, the authors propose a new favorable pathway for the formation of methyl acetate due to the synergistic action of 8MR and 12MR of mordenite. The acylium ion created in the carbonylation of the surface methyl group diffuses to 12MR to undergo a further reaction with MeOH/DME. Products of this reaction are methyl acetate and protonated methanol (CH_3OH_2^+), which diffuses back to 8MR and restores the surface methyl group. This study therefore provides a novel and very interesting perspective on the research subject, as in previous studies the preferred mechanism was traditionally explained via less energetically efficient transformations involving surface acetyl in 8MR, while the formation of C-C bonds in 12MR was attributed only to the formation of coke.

The authors used advanced ab initio molecular dynamics simulations to explore the reactions and effects involved in this novel pathway; however, I have some points that should be addressed by the authors before I can recommend the manuscript for publication:

Author Reply: Thanks for your carefully checking and valuable suggestions to our work, these are very helpful to improve the quality of this work.

1) For simulations in 12MR and 8MR, the authors used two different zeolite models with different positions of Al atoms in the framework. The presence or absence of Al in channels and cavities affects the energetics of reactions and processes that occur in the catalyst. Therefore, a direct comparison of processes such as diffusion, which was studied in the submitted work, might be affected by this.

The model having Al in 8MR has different properties than the model with Al in 12MR, and also a model with two Al atoms in both 8MR and 12MR, which was not considered in this study, but in which it could be assumed that the additional Al atom would be able to stabilize some intermediates. Could the authors comment on to what extent the position of the Al atom affects the results? In addition, could the authors provide more detailed figures of cells with highlighted positions as a part of SI.

Author Reply: Thanks for your professional comments, we have simulated the diffusion process of acylium ion from 8MR channel to 12MR channel in the presence of additional BAS in 12MR channel as

described in **Figure S10**. Based on the obtained free energy profile, the diffusion process was greatly influenced by the additional BAS as the reviewer's comment. However, the additional BAS in the 12MR channel cannot promote the spontaneous MA formation via acylium ion, because the migrated acylium ion will transfer to surface acetyl and the MA formation need to overcome the extra free energy barrier of the reaction between surface acetyl and DME/MeOH as described in **Fig. 2a** and **2b**. These results indicate that Al atom in the 12MR channel of MOR will greatly affect the migration of acylium ion from 8MR channel to 12MR channel, and the elimination to Al atom in the 12MR channel of MOR can not only avoid the C-C bond coupling in MOR-12MR but also be benefit to the MA formation.

We have added the description in the main text and the detailed explanation in Supporting Information.

In the main text:

“Moreover, we further considered the influence of the existence of additional BAS in the 12MR channel to the migration of acylium ion (**Figure S12**), and acylium ion will transfer to surface acetyl but the reaction to MA formation will not spontaneously occur. Therefore, Al atom in the 12MR channel of MOR will greatly affect the migration of acylium ion from 8MR channel to 12MR channel, and the elimination to Al in the 12MR channel of MOR can not only avoid the C-C bond coupling in MOR-12MR but also be benefit to the MA formation.”

In the Supporting Information:

Figure S12. The migration of the acylium ion from the side pocket to the 12MR channel in the presence of additional BAS in 12MR channel, (a) no additional molecule in 12MR channel, (b) one DME in 12MR channel, and (c) one MeOH in 12MR channel.

“Unlike the acylium ion migration without additional BAS in the 12MR channel (**Figure 4**), the migrated acylium ion will transfer to surface acetyl in the presence of additional BAS in the 12MR channel, and the formed surface acetyl will lead to the weak Al-O(H) bond as displayed in **Figure S10a**. Moreover, the introduction of DME/MeOH in the 12MR channel will not lead to spontaneous reaction between acetyl species and DME/MeOH (**Figure S10b** and **S10c**), and the DME/MeOH will be firstly protonated by additional BAS and migrated acylium ion will directly transfer to surface acetyl via the C-O bond coupling between positive charged acylium ion and negative charged AlO_4^- . In this context, the acylium ion completely transfer to the surface acetyl in the 12MR channel and the formation of MA will overcome the extra free barrier as illustrated in **Figure 2a** and **2b**. Overall, the existence of BAS in the 12MR channel is detrimental to the migration of acylium ion and the spontaneous MA formation.”

The cell including surface acetyl and acylium ion in mordenite has been added into Supporting Information:

Figure S1. Periodic structure models of mordenite zeolite in this work, (a) acylium ion in side pocket with Si-O⁻-Al (b) acylium ion in side pocket with Si-O⁻-Al and additional BAS in 12MR channel, (c) surface acetyl in 8MR channel of mordenite, (d) surface acetyl in 12MR channel of mordenite.

2) In the submitted work, as shown in Scheme S1, the authors used a different CV for MOR-8MR in the MTD simulation than for MOR-12MR and SSZ-13 in the case of MA formation via surface acetyl. The choice of a good CV that properly describes the progress of reactions can be a very difficult task. In the work of B. Peters (<https://doi.org/10.1146/annurev-physchem-040215-112215>), the properties and criteria which a good reaction coordinate should fulfill are discussed. It would be useful if the authors commented on their choice of CVs that were used in their work and how they confirmed their quality.

Author Reply: I highly agreed with your comment on the choice the right CV to properly describe the progress of reactions. In this work, the different chemical environment between MOR-8MR and MOR-12MR let us chose the different CV to describe the MA formation. In MOR-8MR, surface acetyl cannot directly react with methanol to form MA because of the strong confinement effect of side pocket to both surface acetyl and methanol, therefore, C-O bond between acetyl and Si-O-Al are required to firstly

dissociate to acylium ion, and then acylium ion can bond to methanol for the formation of MA as displayed in **Fig 3c**. In contrast, the MA formation in MOR-12MR and SSZ-13 cannot use the same CV to describe, there are two reasons, 1) the driving force of MA formation between surface acetyl and methanol (dimethyl ether) in the open space should be not the automatic activation of surface acetyl but the inducing effect of methanol (dimethyl ether) to surface acetyl; 2) the C-O bond of surface acetyl in MOR-12MR and SSZ-13 will be easily broken with the bias potential on this bond, and the formed acylium ion will quickly deprotonate to ketene as confirmed by our previous study, this side reaction is not what we want to avoid in the MA formation. In summary, the different chances to CVs as reaction coordinates in MOR-8MR and MOR-12MR (SSZ-13) is exactly to better describe the reaction after many attempts, and our current setting to CVs in all models fulfill three requirements to reaction coordinates proposed by Peters.

We have added detailed descriptions to choose these CVs in chapter **1.2 Enhanced sampling methods** of Supporting Information:

“Herein, the different CVs between MOR-8MR and MOR-12MR were chosen to well describe the MA formation in different confined spaces. In MOR-8MR, methanol cannot well induce the C-O bond rupture of surface acetyl due to the limitation of the narrow space of side pocket and 8MR channel, therefore, the bias potential was performed on the coordination number of C-O bond in surface acetyl for the formation of acylium ion firstly, and then acylium ion with the high mobility can bond to methanol for the formation of MA. But in MOR-12MR, the inducing effect of methanol or DME to surface acetyl can play their full role to form MA, and the bias potential to the coordination number of C-O bond in surface acetyl will lead to the formation of acylium ion and further deprotonate to ketene as the side reaction. In summary, the different chances to CVs as reaction coordinates in MOR-8MR and MOR-12MR is exactly to better describe the reaction after many attempts, and our current setting to CVs in all models fulfill the requirements to reaction coordinates proposed by Peters.^{1”}

3) *The authors state that the formation of surface acetyl in 8MR is hindered and energetically unfavorable, however, in the work of M. Gešvandtnerová et al. (<https://doi.org/10.1016/j.jcat.2021.02.011>) surface acetyl was discovered from molecular dynamics simulation of acylium ion after some time in the side pocket. This suggests that the formation of surface acetyl in 8MR is energetically feasible, and the barrier for the conversion can be overcome even without treating the transformation like a rare event with a need for specialized MD techniques. This makes me wonder how did the authors know when to stop the equilibration period of their simulations, and how did the authors make sure that their simulations are long enough?*

Author Reply: I am sorry to the raised misunderstandings, the formation of surface acetyl from acylium ion in 8MR channel is energetically favorable and hindered by two small free energy barriers as displayed in **Fig. 4**, and the free energy profile of the conversion between surface acetyl in 8MR channel and acylium ion in side pocket was obtained from our previous work (*J. Am. Chem. Soc.* **2021**, 143, 15440–15452).

We have added the sentence to indicate the origin of this free energy profile in the caption of Fig. 4.

“The free energy profile of the interconversion between surface acetyl and acylium ion (SP) was reproduced based on our previous results.^{26”}

In this work, our AIMD simulations of acylium ion in this work mainly focus on its mobility from side pocket to 12MR channel with or without the presence of MeOH/DME, and no simulation about their interconversion was carried out. But we note that the AIMD simulation to acylium ion in the side pocket has confirmed that acylium ion will not deprotonate to ketene nor transfer to surface acetyl after 50 ps regular AIMD simulations (*J. Am. Chem. Soc.* **2021**, 143, 15440–15452), in contrast, the AIMD simulation of M. Gešvandtnerová et al. (*J. Catal.* **2021**, 396, 166–178), acylium ion will transform to surface acetyl

in the side pocket of mordenite after > 90 ps. Based on the flat free energy profile between surface acetyl and acylium ion in **Fig. 4**, surface acetyl can also transform to acylium ion after a longer timescale. However, the limitation of computational source prevents us explore the dynamic evolution between surface acetyl in 8MR channel and acylium ion in side pocket.

Herein, we have added solid state NMR (ssNMR) experiments of acetyl chloride ($\text{CH}_3^{13}\text{COCl}$) adsorption in mordenite to confirm the existence of acylium ion and the interconversion between surface acetyl and acylium ion (**Fig. 5**). We can observe that $\text{CH}_3^{13}\text{COCl}$ will firstly dechloridate to surface acetyl (183 ppm) by the BAS in 12MR channel and further deprotonate to ketene with high mobility, and the formed ketene will further diffuse into 8MR channel to form surface acetyl (187 ppm) or acylium ion (169 ppm). *in situ* NMR spectra at different time of heat treatment (333 K) shows the dynamic evolution among surface acetyl in 12MR channel, surface acetyl in 8MR channel, and acylium ion in side pocket. Consistently with the thermodynamical stability and low barrier between surface acetyl and acylium ion, the signal intensity of surface acetyl (187 ppm) in 8MR channel is much higher than acylium ion (169 ppm) in side pocket. Therefore, the dynamic equilibrium between surface acetyl and acylium ion actually exists, and the reaction mechanism of MA formation via migrated acylium ion is reasonable.

“Solid state NMR of acetyl species in mordenite.

To confirm the above-mentioned mechanism, the solid-state NMR (ssNMR) experiments of acetyl chloride ($\text{CH}_3^{13}\text{COCl}$) adsorption on mordenite zeolite were employed to understand the interconversion of surface acetyl and acylium ion in mordenite zeolite as displayed in **Fig. 5**. The acetyl chloride will rapidly transform to surface acetyl with the dechloridation of BAS in MOR-12MR ($\text{CH}_3^{13}\text{COCl} + \text{Zeo-OH} \rightarrow \text{Zeo-O}^{13}\text{COCH}_3 + \text{HCl}$) at 298 K when the partial pressure of $\text{CH}_3^{13}\text{COCl}$ is 0.5 kPa, and the signal of 183 ppm and 187 ppm can be assigned to the surface acetyl in 12MR channel and 8MR channel based on the reported 185 ppm by Li *et. al.*⁴⁰ The increasing concentration of acetyl chloride will further lead to the appearance of 167, 172, and 177 ppm, and 172 and 177 ppm can be assigned to unreacted acetyl chloride with the physical adsorption and hydrogen bond adsorption in 12MR channel based on the relationship of acetyl chloride concentration and signal strength, but the signal of 167 ppm can be attributed to acylium ion in side pocket based on our calculated value of 168 ppm. Due to the lower thermodynamical stability of acylium ion than surface acetyl (**Fig. 3b**), the signal strength of acylium ion is much lower than surface acetyl and acylium ion was formed after the formation of surface acetyl.

It's worth noting that the acetyl chloride cannot directly react with the BAS in 8MR channel of mordenite zeolite due to the size exclusion, but the appearance of 187 ppm (surface acetyl in 8MR channel) indicates the existence of transferred mediator between 8MR and 12MR channels, *i.e.*, linear ketene (CH_2CO). Herein, acetyl chloride was firstly dechloridated to surface acetyl in 12MR channel, and then surface acetyl was deprotonated to ketene with high mobility and further diffuse into the 8MR channel, and finally ketene was protonated to surface acetyl in 8MR channel or acylium ion in side pocket by BAS in 8MR channel (**Fig. 5d**). To confirm this process, we further employed ssNMR to explore the evolution of acetyl species as displayed in **Fig. 5b**, the strength of 183 ppm (surface acetyl in 12MR channel) will be weakened with the increasing time of heat treatment at 333 K but the signals of surface acetyl in 8MR channel (187 ppm) and acylium ion in side pocket (167 ppm) are enhanced. All these results confirm our hypothesis in **Fig. 5c** and the relative thermodynamical stability and dynamic interconversion between acylium ion and surface acetyl in MOR-8MR (**Fig. 4b**).

Fig. 5. (a) ^{13}C MAS NMR spectra of different loadings of $\text{CH}_3^{13}\text{COCl}$ reacted with dehydrated H-MOR zeolite at 333 K for 2 h. (i) 0.5 kPa, (ii) 0.8 kPa, (iii) 1.0 kPa, and (iv) 1.5 kPa. (b) *in situ* ^{13}C MAS NMR spectra of 0.5 kPa $\text{CH}_3^{13}\text{COCl}$ adsorbed on dehydrated H-MOR zeolite at different reaction conditions. (c) Assignment to chemical shifts. (d) relationship between surface acetyl and acylium in 12MR and 8MR channels.”

4) As a part of the main text, the authors provide a section “Methods” with a subsection “Theoretical Calculations” in which the authors refer the reader to their previous work for computational details. This section should be extended, or can be added as a part of SI, with relevant computational settings and details, such as the particular DFT functional, convergence criteria and length of their simulations for MTD and for US windows.

Author Reply: The details of computational settings have been replenished for clarity in the main text.

“However, the non-spontaneous reaction was explored by MTD method employed for enhancing the probability of sampling chemical reactions or rare events provided that a limited number of collective variables (CV) describing the reaction coordinates were well-defined as illustrated in **Scheme S1**. Nonetheless, MTD simulation is normally biased by regularly spawning Gaussian hills along the chosen CVs, which was defined by coordination numbers (CN):

$$\text{CN} = \sum_{ij} \frac{1 - \left(\frac{r_{ij} - d_0}{r_0}\right)^n}{1 - \left(\frac{r_{ij} - d_0}{r_0}\right)^m}$$

where r_{ij} represents the interatomic distance between atoms i and j and r_0 is the reference distance. In this work, parameters d_0 , n , and m were set to 0, 6, and 12, respectively. The 2D free energy profile was reproduced according to the bias potential of MTD algorithm implemented within PLUMED, that is defined as:

$$F(S) = -k_B T \left(\int dR \delta(S - S(R)) e^{-\frac{U(R)}{k_B T}} \right)$$

k_B is the Boltzmann constant, T is the temperature of the system, $U(R)$ is the potential energy function, and $S(R)$ is the CVs.

Gaussian hill width and height of all MTD simulations were summarized in **Table S2**, MTD simulations were allowed to continue till the height of additional hills no longer affect the resultant free energy profile, and the MTD simulations were considered to be converged if the barriers between every 500 hills added do not differ by more than 1 kJ/mol as displayed in **Figure S18**. At least 50 ps AIMD-MTD simulations were used to fully guarantee the reaction equilibrium. Based on the sum of spawned Gaussian hills, the 2D free energy profile of the reaction may be reconstructed. Accordingly, a lowest free energy path (LFEP) was derived by means of the MEPSA software.⁵⁰

In the US, the mobilities of the reactant species from the 8MR channel to the 12MR channel were defined by a collective variable, (the distance between the two centers of mass (**Schemes S2–S4**)), that is, the reactant species and the 8MR window. The additional potential introduced by a simple harmonic term:

$$U(s) = \frac{k}{2}(x - x_0)^2$$

Where x is the real-time value of CV, x_0 is the value of specified value of CV, k is set to 50 kJ/mol for acylium ion, DME, MeOH, and MA. All biased umbrella sampling simulations were consisted 10^4 steps at NVT ensemble, and the setting to k lead to the reasonable sampling indicated by the inter-overlapping of different windows. The free energy profile was generated by the sampling obtained in each window based on the weighted histogram analysis method.⁵¹ The error bars of free energy profiles generated from MTD and US simulations have been calculated as displayed in **Figure S19** and **S20**, and more details of error bars were illustrated in Supporting Information.”

5) *The authors came to their conclusions by comparing the energy spans calculated from the free energies obtained from the free energy profiles for reactions consisting of multiple steps. Energy of each step is usually calculated with some error, which is cumulated in the resulting energy span. Thus, I am interested in what accuracy was achieved for the results and if the energy difference is still statistically significant even when errors are included. Could the authors comment on the accuracy of their energy spans and their estimation of errors?*

Author Reply: Thanks for your professional comments, we have performed the calculation to error bars of MTD based on Nastase et al. (*ACS Catal.* **2020**, 10, 8904-8915) and US simulations based on weighted histogram analysis method (*Comput. Phys. Commun.* **2001**, 135, 40-57), and the calculated details have been added to Supporting Information.

“1.3 Error bars of free energy in MTD and US simulations

In order to determine the error bars, the simulation was continued such that a further 1000 hills were added to the system after convergence. The error bar with respect to the transition state was then calculated as the average between the energy barrier at the moment of convergence and the energy barrier one these 200 additional energy hills had been included (**Figure S19**).² The error bars of free energy profile in US simulation were directly generated with free energy profile by weighted histogram analysis method (**Figure S20**).³”

To evaluate the errors of the free energy profile, the representative reaction of surface acetyl + MeOH to MA in MOR-8MR has been used to calculate the error bar of MTD simulations as an example (**Figure S18** and **S19**), and the diffusion process of acylium ion from 8MR channel to 12MR channel has been employed to assess the errors of US simulations (**Figure S20**). Obviously, the error bars of both MTD and US simulations were smaller than ± 1.2 kJ/mol, and the multiple steps of free energy profile will lead to a total error smaller than 2 kJ/mol. Therefore, the obtained free energy profiles from both MTD and US simulations are converged and methodologically accurate.

Figure S18. Convergence test of lowest free energy path (LFEP) in the reaction of surface acetyl + MeOH to MA in MOR-8MR. Notably, the free energy barrier is converged when the simulated time is 35 ps according to the same maximum value at the transition state, and LFEP is converged when the simulated time is longer than 45 ps from the perfectly coincident lines between 45 ps and 50 ps.

Figure S19. The lowest free energy path (LFEP) in the reaction of surface acetyl + MeOH to MA in MOR-8MR with the different MTD simulated time to estimate the error bar of free energy barrier after reaction equilibrium.

Figure S20. Free energy profile with error bar of the movement of the acylium ion into the 12MR channel previously occupied by DME and the further formation of MACH_3^+ , *i.e.*, the described process in **Fig. 5b**.

The authors also consider the interconversion of surface acetyl to acylium ion, with surface acetyl being energetically lower (shown in Fig. 4). Calculating the energy spans of two competitive reactions is appropriate if they share the same most stable determining intermediate. Therefore, the energy span for the right side of the profile (Fig. 4) should be calculated with surface acetyl as a determining intermediate, as it is energetically lower, and therefore it maximizes the energy span, as described in the work of Kozuch *et al.* (<https://doi.org/10.1021/ar1000956>). Energy span for the right side of the profile in Fig. 4 calculated this way is comparable to the energy span for the left side of the profile. Taking into account the possible errors, these values could be equivalent, and then it is questionable if such a clear final conclusion can be made.

Author Reply: Yes, using energetically more favorable surface acetyl as the reference state to obtain energy span is more aligned with the criterion of Kozuch *et al.* in *Acc. Chem. Res.* **2011**, 44, 101–110, and **Fig. 4** has been updated using surface acetyl as the reference state for both right and left sides. Note that the errors of free energy profile in **Fig. 4** have been confirmed to be smaller than ± 1 kJ/mol, and the determination to the more favorable pathway of MA formation based on this free energy profile is still credible. Based on the free energy profile in **Fig. 4**, the formation of methyl acetate *via* the migrated acylium ion as an intermediate in 12MR channel using both DME and MeOH as reactants (right side) are still kinetically more favorable than that *via* the surface acetyl and MeOH in 8MR channel (left side).

However, we noted that energy span (140.2 kJ/mol) of MA formation *via* acylium ion and 2 DME are very close to that *via* the surface acetyl and MeOH in 8MR channel (142.3 kJ/mol) but much larger than that *via* acylium ion and 2 MeOH (or MeOH + DME) (105.2 kJ/mol and 115.0 kJ/mol). Combined the forbidden pathway of MA formation between DME and surface acetyl in MOR-8MR (**Fig. 1c**), these results not only describe the priority of MA formation *via* the migrated acylium ion but also emphasize the influence of different reactants to MA formation, and such important new conclusion has been added to main text.

“Notably, the high DME concentration (2DME) would also result a large δE (140.2 kJ/mol) of MA formation even using acylium ion as the intermediate, but the pathway of MA formation *via* acylium ion migration is thermodynamically more favorable than that *via* surface acetyl as displayed in **Fig. 3b**. Overall, DME is not a beneficial reactant, these results not only describe the priority of MA formation *via* the migrated acylium ion but also emphasize the influence of different reactants to MA formation, and the more important role of DME as a reactant in DME carbonylation might be to generate SMS and MeOH to facilitate the C-C bond coupling in MOR-8MR and MA formation in MOR-12MR, respectively.”

Fig. 3. MA formation based on the acylium ion in the side pocket (SP) and surface acetyl in the 8MR channel. (a) reaction pathways, and (b) free energy profile. The dotted lines in (b) indicate the diffusion process; the values in black are the free energy barriers; the values in pink are the maximum free energy spans. The free energy profile between surface acetyl and acylium ion (SP) was reproduced based on our previous results.²⁵

6) This point follows up the previous point concerned with the accuracy of the calculated free energies. From the computational details that are provided, it is not clear to me how the authors took into account the terms of the probability densities of their reactant configurations and velocities in the transition states.

Here, I am referring to Eq. 7 in the work of M. Gešvandtnerová et al. (<https://doi.org/10.1016/j.jcat.2021.02.011>), and to the works of E.A. Carter et al. ([https://doi.org/10.1016/S0009-2614\(89\)87314-2](https://doi.org/10.1016/S0009-2614(89)87314-2)), and T. Bučko et al. (<https://doi.org/10.1039/C7CP05562E>).

Author Reply: Thanks for your professional comments, the free energy profile of MTD simulation was generated by PLUMED software based on the Eq of $F(S) = -k_B T \left(\int dR \delta(S - S(R)) e^{-\frac{U(R)}{k_B}} \right)$ as defined by Barducci et. al. in *Wiley Interdiscip. Rev.: Comput. Mol. Sci.* **2011**, 10.1002/wcms.31. This equation has been added to computational details in main text:

“The 2D free energy profile was reproduced according to the bias potential of MTD algorithm implemented within PLUMED, that is defined as:

$$F(S) = -k_B T \left(\int dR \delta(S - S(R)) e^{-\frac{U(R)}{k_B}} \right)$$

k_B is the Boltzmann constant, T is the temperature of the system, $U(R)$ is the potential energy function, and $S(R)$ is the CVs.”

Due to the different enhanced sampling methods between your mentioned work and this work, blue moon ensemble method and metadynamic method, respectively, it's hard to evaluate the probability density of the reactant configurations and velocities in the transition states using the same strategy to your mentioned work. Herein, the probability distribution functions of approximation to the CV1 in the MTD simulation of surface acetyl + MeOH to MA + BAS in MOR-8MR were used to evaluate the probability density of the reactant configurations, and the high probability of the reference reactant states (CV1 = -1.03) indicates the sufficient sampling to reactants as displayed in **Figure S21**. Moreover, the radial distribution function for the atoms pair of C_{acetyl} and O_{Zeo} during the MTD simulation of surface acetyl + MeOH to MA in MOR-8MR was used to evaluate the sufficiency of sampling to transition state as displayed in **Figure S22**, and the efficient and reasonable distribution of $d(\text{C}_{\text{acetyl}}\text{-O}_{\text{Zeo}})$ at the transitional range of bond rupture (1.70 Å ~ 2.00 Å) indicate the enough sampling to transition state. In summary, the sufficient sampling to both reactant and transition state guarantee the accuracy of free energy profiles. The details of sampling analysis have been added to Supporting Information:

“To evaluate the sufficiency of bias sampling in MTD simulation, the probability distribution functions of approximation to the CV1 in the MTD simulation of surface acetyl + MeOH to MA + BAS in MOR-8MR were used to evaluate the probability density of the reactant configurations, and the high probability of the reference reactant states (CV1 = -1.03) indicates the sufficient sampling to reactants as displayed in **Figure S21**. Moreover, the radial distribution function for the atoms pair of C_{acetyl} and O_{Zeo} during the MTD simulation of surface acetyl + MeOH to MA in MOR-8MR was used to evaluate the sufficiency of sampling to transition state as displayed in **Figure S22**, and the efficient and reasonable distribution of $d(\text{C}_{\text{acetyl}}\text{-O}_{\text{Zeo}})$ at the transitional range of bond rupture (1.70 Å ~ 2.00 Å) indicate the enough sampling to transition state. In summary, the sufficient sampling to both reactant and transition state guarantee the accuracy of free energy profiles.”

Figure S21. Probability distribution functions of approximation to the CV1 determined using the MTD simulations for the reactant configurations (surface acetyl + MeOH in MOR-8MR). Dashed lines indicate the reference reactant states used in the free energy calculations.

Figure S22. Radial distribution functions ($g(r)$) for the atoms pair of C_{acetyl} and O_{Zeo} during the MTD simulation of surface acetyl + MeOH to MA in MOR-8MR. The highlighted region ($1.70 \text{ \AA} \sim 2.00 \text{ \AA}$) can be approximately considered as the transition state used in the free energy calculations.

Minor points:

7) Is there a reason why free energy profiles are called ‘free energy surface’ and vice versa? The profile in Fig. 1c is denoted in the caption as ‘surface’ and the free energy surfaces in Fig. 2 are denoted as ‘free energy profiles’. This point is valid also for Figures 3, S1, S3, S5, S9, S10 and S11.

Author Reply: I am sorry to this inconsistent expression because of carelessness, all “free energy surface” have been corrected to “free energy profile”

8) Movies are mentioned in the main text, but no movies were provided in the zip file for reviewers.

Author Reply: The Movies have been supplied in the new submission.

9) The color code in Fig. 4 is not properly chosen – the pink values are not pink, but purple, however the lines are pink. This is a bit chaotic.

Author Reply: The color code in Fig. 4 has been updated for clarity.

10) The caption for Fig. 8 is missing (iii) – (v).

Author Reply: The missing caption of (iii)-(v) in Fig. 8 has been replenished.

Reviewer #2 (Remarks to the Author):

This manuscript deals with investigations of the reaction mechanism of CO methylation in the zeolite Mordenite using mostly molecular dynamics simulations but also IR spectroscopy. Based on the MD data new pathways and intermediates are suggested for the formation of methyl acetate including previously identified molecules such as ketene.

Overall, I find this study very specific, even for zeolite catalyzed reactions, as this is investigating one key step for one specific zeolite morphology. I hence do not see how this manuscript should be of vivid interest to the broad audience of Nature Communications, nor do I see the "smoking gun" that makes the findings here a hot topic. I therefore recommend to reject this manuscript and suggest resubmission to a catalysis specialized journal, e.g. Journal of Catalysis.

Author Reply: Thanks for your comments, I am sorry to not well illustrate the importance and significance of current work. To highlight our important finding in this work, (i) the importance of carbonylation in mordenite in catalysis and chemical industry was emphasized in the introduction part, (ii) the writing was greatly simplified to attract the broad interest of *Nature Communications* by rewriting and reorganizing, (iii) solid state NMR experiments were added to further confirm our proposed mechanism, (iv) additional calculations to the diffusion of acylium ion in the presence of additional Al distribution provide a solid evidence that the proposed mechanism is widely existed in carbonylation.

Notably, the proposed intermediate migration mechanism should widely exist in zeolite catalyzed reactions. Analogously to acylium ion, the carbocation as one kind of common intermediate in zeolite catalyzed reactions (like methanol to olefins, alkene cracking, alkylation, *etc.*) also has the strong mobility, especially at high reaction temperature, and the mobilizable carbocation in zeolite confined environment can greatly interact to some inaccessible active sites. (*Nature*, **1993**, 363, 529-531; *J. Am. Chem. Soc.* **2015**, 137, 32, 10374–10382; *Angew. Chem. Int. Ed.* **2022**, 61, e202111180.) The charge separation between carbocation and AlO_4^- lead to the high activity of carbocation kinetically, and the confined environment of zeolite framework can help to stabilize carbocation in an active state thermodynamically.

Moreover, the reaction mechanism in this work was proposed by an integrated reaction/diffusion kinetics model, and this model was never reported by previous studies but of great necessity to the study of reaction mechanism in zeolite catalysis. For example, zeolites, as an indispensable platform of multi-functional catalysts in syngas conversion and CO_2 reduction, are widely employed to overcome the product selectivity limitations and improve the catalyst stability, and the shape selectivity of zeolite is decided by both reactivity and mass transport but is far from full understanding since previous computational studies considered the diffusion and reaction separately. The integrated reaction/diffusion kinetics model proposed by this work can be transplanted into other reaction in zeolites (like methanol to olefins (MTO)) to produce the free energy landscape of complete catalytic cycle, therefore, the reaction mechanism of MTO will be fully understood from the perspective of both reaction and diffusion.

Therefore, the proposed intermediate migration mechanism should be not limited on the carbonylation in mordenite, and the other reaction process in zeolite catalyzed reaction with confinement

may be re-evaluated to draw a conclusion than before by the proposed integrated reaction/diffusion kinetics model. Considering all these efforts on supporting the proposed mechanism and great improvements on broad interest, please help to re-evaluate the possibility of this work publication on *Nature Communications*.

We have added some descriptions in the main text:

In Abstract:

“The synergistic effect between different channels appears to account for the high carbonylation activity in mordenite that has thus far not been fully explained, and this paradigm may be used to rationalize the observed catalytic activity of other reactions involving multiple-porous and structural confinement effects.”

In Introduction:

“Carbonylation in zeolites provides a convenient route and platform in both zeolite catalysts and multi-functional catalysts to functionalize C1 intermediates *via* formation of carbon-carbon bonds at relative low temperatures to realize syngas conversion,² CO₂ reduction,³ and methanol to olefins.⁴”

In Discussion:

“Notably, the proposed intermediate migration mechanism can also be generalized to other zeolite catalyzed reactions. Analogously to acylium ion, the carbocation as one kind of common intermediate in zeolite catalyzed reactions (like methanol to olefins, alkene cracking, alkylation, *etc.*) also has the strong mobility, especially at high reaction temperature, and the mobilizable carbocation with the charge separation between carbocation and AlO₄⁻ lead to the high reactivity of carbocation kinetically, and the confined environment of zeolite framework can help to stabilize carbocation thermodynamically. Additionally, this new mechanism was proposed by an integrated reaction/diffusion kinetics model, and all processes (adsorption & desorption, reaction, and diffusion) of the complete catalytic cycle were involved in a free energy landscape, which makes the integrated consideration to reaction mechanism possible. Such model can also be applied to understand the product selectivity of zeolites in the reaction of MTO, syngas conversion, and CO₂ reduction.”

Reviewer #3 (Remarks to the Author):

Wei et al. investigated methyl acetate (MA) formation via different intermediates in MeOH/DME carbonylation over mordenite and SSZ-13 through AIMD simulations using enhanced sampling methods. With the high activity and mobility of the acylium ion in the side pocket (SP) of mordenite, it is more thermodynamically and kinetically favorable for MA formation via acylium ion than the traditional pathway with surface acetyl. With employing advanced ab initio molecular dynamic simulations and two-dimensional correlation analysis of rapid scan infrared spectroscopy, this work has proposed a new pathway for MA formation via acylium ion in the 12MR channel. In this pathway, the acylium ion in SP can easily migrate into MOR-12MR and spontaneously form MA. This newly proposed mechanism acts through 3 pore architectures acting synergistically in differently confined environments to promote the efficient carbonylation in mordenite via mobile acylium ion intermediate. It can be confirmed with the direct spectroscopic evidence that the 8MR channel plays a key role in MA formation and the migration of reactive species to the 12MR channel can also be found through 2-D COS FTIP analysis. However, there are major concerns about the theoretical work for possible publication. The detailed comments are provided as following:

Author Reply: Thanks for your careful reviewing and high evaluation to our work, and your suggestions

are really helpful to improve the quality and specialty of this work.

1. Fig.1d showed the energy decomposition analysis between organic species and zeolite framework. However, the authors didn't clearly explain how they defined and calculated E_{Pauli} , E_{elec} , E_{orb} , E_{disp} and E_{total} . More comprehensive explanations are needed here. In addition, the motivation for authors to compare these parameters are also needed here.

Author Reply: Thanks for your insightful comments, the energy decomposition analysis was used to quantitatively compare the repulsion of different zeolite framework to guest molecule in TS 3 (**Fig 1c**) more details of energy decomposition analysis have been added to Supporting Information, and the reason of energy decomposition analysis has been described in the main text.

In the main text:

“To explain such high free energy barrier, the energy decomposition analysis was used to quantitatively compare the repulsion of different zeolite framework to guest molecule in TS3, and an examination of the Pauli repulsion between reacting species in the MOR-8MR and MOR-12MR channels further confirms the much higher steric strain present in the narrower 8MR channels (**Fig. 1d, 1e, and 1f**).”

In the Supporting Information:

“1.4 Energy decomposition analysis

Energy decomposition analysis (EDA) was used to quantitatively describe the host-guest interaction of TS3 (Fig. 1d) in different channels of MOR zeolite. Detailed and precise analyses of host-guest interactions may be accomplished based on the ETS-NOCV EDA approach,⁴ which combined the extended transition state (ETS) method with the natural orbital for chemical valence (NOCV) theory by means of the Amsterdam density functional (ADF) program.⁵ The PBE-D3 functional was chosen for the DFT calculations and EDA to warrant the uniformity between dynamic and static calculations. By invoking the frozen core approximation, a double-zeta Slater type orbital (STO) basis set containing polarization functions (namely, DZP) was adopted for all elements to describe interactions between intermediates and the zeolite framework.⁶ Auxiliary STO functions, centered on all nuclei, were used to fit the electron density and to obtain accurate Coulomb potentials in each SCF cycle. For the ETS-NOCV EDA scheme, the interaction energy (ΔE_{total}) between the fragments may further be divided into four components:

$$\Delta E_{\text{total}} = \Delta E_{\text{Pauli}} + \Delta E_{\text{elec}} + \Delta E_{\text{orb}} + \Delta E_{\text{disp.}}$$

Where the four energy terms respectively account for the Pauli repulsive interaction among occupied orbitals on the fragments (ΔE_{Pauli}), the classical electrostatic interaction between two fragments (ΔE_{elec}), electron distribution of the constituent molecules (ΔE_{orb}), the dispersion interaction ($\Delta E_{\text{disp.}}$) due to the use of dispersion corrected PBE-D3 functional.”

2. The author applied MA formation over SSZ-13 mechanism for comparison with that over MOR. But SSZ-13 and MOR have many different factors that could lead to the different catalytic performance of reactions, i.e., confinement, membrane size, transport limitation, acid/base site density. Please clarify.

Author Reply: Thanks for your comments, we highly agreed your points that many different factors that can lead to the different catalytic performance of reactions, i.e., confinement, membrane size, transport limitation, acid/base site density. However, these factors cannot be well reflected by calculations. To strongly emphasize our proposed mechanism in mordenite zeolite of this work, we decided to delete all results of SSZ-13 in this work, and the revised manuscript is briefer and clearer than before. Surely, the mentioned factors by referee is worth to highlight in the main text, and we have added some descriptions to strength the importance of confinement, membrane size, transport limitation, acid/base site density of

zeolites to catalytic performance

“Additionally, many other factors of zeolite can also greatly influence the catalytic performance of carbonylation, like membrane size and thickness, acidic site density, surface barriers, *etc.*”

3. *More motivation details are needed for Fig. 2 to employ the coordination number for illustrating the reaction mechanism.*

Author Reply: we have added the motivation details of coordination number in Supporting Information. In caption of Fig. 2:

More motivation details of CN setting can be found in Supporting Information.

In Supporting Information:

“Herein, the different CVs between MOR-8MR and MOR-12MR were chosen to well describe the MA formation in different confined spaces. In MOR-8MR, methanol cannot well induce the C-O bond rupture of surface acetyl due to the limitation of the narrow space of side pocket and 8MR channel, therefore, the bias potential was performed on the coordination number of C-O bond in surface acetyl for the formation of acylium ion firstly, and then acylium ion with the high mobility can bond to methanol for the formation of MA. But in MOR-12MR, the inducing effect of methanol or DME to surface acetyl can play their full role to form MA, and the bias potential to the coordination number of C-O bond in surface acetyl will lead to the formation of acylium ion and further deprotonate to ketene as the side reaction. In summary, the different chances to CVs as reaction coordinates in MOR-8MR and MOR-12MR is exactly to better describe the reaction after many attempts, and our current setting to CVs in all models fulfill the requirements to reaction coordinates proposed by Peters.¹”

4. *On Page 11, the authors said that the surface acetyl and acylium ions are the possible intermediates connecting C-C bond coupling and MA formation in MOR-8MR. However, the following part gave an assumption that surface acetyl is the sole reacting species. Please strengthen this assumption with experimental evidence. Also, why not take the acylium ions into consideration in this part?*

Author Reply: Thanks for your comments, the interconversion between surface acetyl and acylium ion is very easy to realize as seen from the flat free energy profile in **Fig. 4b** (black line), but surface acetyl is more stable than acylium ion. Therefore, we preferentially considered the MA formation in 8MR via surface acetyl as displayed in **Fig. 3**, and then acylium ion was considered as the other intermediate to move into 12MR channel as displayed in **Fig. 5**. The details of free energy profile of MA formation using two intermediates have been illustrated in **Fig. 4b**, and the left side is surface acetyl and the right side is acylium ion. To further clearly illustrate our results, the reaction mechanism of MA via surface acetyl in both MOR-12MR and MOR-8MR has been organized as a new chapter with the sub-title of **Methyl acetate formation via surface acetyl**, and the reaction mechanism of MA formation via acylium ion has been organized as the other chapter with the sub-title of **MA formation via migrated acylium ion**. Moreover, the interconversion between surface acetyl and acylium ion in mordenite has been confirmed by our NMR experiment in **Fig. 5**, and we have added a new chapter of **Solid state NMR of acetyl species in mordenite** in the main text.

5. *The selectivity of zeolite materials is influenced by the channel sizes and molecule sizes. Seen in Fig. 7, a new mechanism for MA formation with acylium ions migration has been provided. Is that possible for acylium ions bonding with surface methyl species in 8MR or 12MR side rather than in SP?*

Author Reply: Thanks for your comments, in this work, MA formation via acylium ion and MeOH/DME occurred in two processes, one is MA formation via the migrated acylium ion and MeOH/DME in 12MR

channel, the other is MA formation *via* the surface acetyl in 8MR channel and MeOH/DME in side pocket and acylium ion is an intermediate. The mentioned surface methyl species are not discussed in this work, and the reaction of two positive charged species may be impossible.

However, your comments mentioned us to compare the reaction pathway of MA formation in 8MR channel, 12MR channel, and side pocket *via* surface acetyl, and we have calculated the free energy profile of MA formation in SP *via* surface acetyl and DME in **Figure S2**. We firstly checked the possible oxygen sites that can bind surface acetyl, and oxygen of T1-O2-T1 and T1-O3-T3 can bind to surface acetyl with the orientation face to side pocket. However, the Al on T1 has been confirmed that the low reaction activity on C-C bond coupling between surface methyl and CO by Corma *et al.* (*J. Am. Chem. Soc.* 2008, **130**, 16316-16323), and only T1-O3-T3 can be used to bind surface acetyl for MA in side pocket. Herein, (Si)T1-O3-T3(Al) was employed to bind surface acetyl as the reactant to form MA. Notably, this surface acetyl on (Si)T1-O3-T3(Al) can be shifted from the surface acetyl on (Si)T3-O3-T3(Al), therefore, the following pathway of MA formation in side pocket can directly connect to the C-C bond coupling in 8MR channel as displayed in new **Fig. 1c**. As displayed in **Figure S2b**, the free energy barriers of MA formation *via* surface acetyl and DME in 8MR channel, 12MR channel, and side pocket are 249.6 kJ/mol, 52.5 kJ/mol, and 164.1 kJ/mol, respectively. Although the free energy barrier in side pocket is greatly decreased relative to that in 8MR channel, this free energy barrier (164.1 kJ/mol) of MA formation in side pocket is greatly larger than the free energy barrier (87.0 kJ/mol) of C-C bond coupling in 8MR channel (**Fig. 1c**). Notably, the free energy barrier of MA formation in side pocket is contributed by the unfavorable adsorption free energy (64.3 kJ/mol) and intrinsic activation energy (99.8 kJ/mol), which indicated that the side pocket with the larger space than is still not enough to allow the MA formation in side pocket. In summary, the MA formation *via* the surface acetyl and DME in both side pocket and 8MR channel is energetically unfavorable, and an alternative reaction pathway is still required to be proposed, *i.e.*, the migrated acylium ion mechanism in this work.

Based on above results, new **Fig. 1** and some descriptions have been added to main text, and Figure S1 have been added to Supporting Information.

In main text:

“Notably, although the free energy barrier in side pocket is greatly decreased relative to that in 8MR channel, this free energy barrier (164.1 kJ/mol) of MA formation in side pocket is greatly larger than the free energy barrier (87.0 kJ/mol) of C-C bond coupling in 8MR channel (**Fig. 1c**). Herein, the free energy barrier of MA formation in side pocket is contributed by the unfavorable adsorption free energy (64.3 kJ/mol) and intrinsic activation energy (99.8 kJ/mol), which indicated that the side pocket with the larger space than is still not enough to allow the MA formation in side pocket.”

Figure S2. (a) T sites and oxygen of mordenite zeolite to bind surface acetyl in MOR-12MR (T4-O10-T4), MOR-8MR (T3-O8-T3), and MOR-SP (T3-O3-T1), (b) free energy profile of MA formation via surface acetyl and DME in MOR-12MR, MOR-8MR, and MOR-SP, (c) geometrical structure of TS3b in MOR-12MR, MOR-8MR, and MOR-SP.

Fig. 1. (a) Pore and channel sizes of mordenite, kinetic sizes of methanol, dimethyl ether, and methyl acetate. The 8MR window refers to the opening between the 12MR channel and the 8MR side pocket (SP) of mordenite. (b) Catalytic cycle of DME carbonylation, the red arrows indicate the rate-determining step, and the green arrows indicate the fast step. (c) Free energy profile over MOR-8MR, MOR-12MR, and MOR-SP at 473 K obtained using the PBE-D3/dgdzvp method. (d) Energy decomposition analysis between organic species and zeolite framework at TS3 in MOR-8MR and MOR-12MR. Host-guest interactions of TS3 in the zeolite framework visualized using the reduced density gradient: (e) MOR-8MR, including both 8MR and SP; and (f) MOR-12MR. Red: repulsion, green: attraction.